# QiMeng-MuPa: Mutual-Supervised Learning for Sequential-to-Parallel Code Translation

**Changxin Ke**[1,2] **Rui Zhang**[1] **Shuo Wang**[1,2] **Li Ding**[2,3] **Guangli Li**[1] **Yuanbo Wen**[1]
**Shuoming Zhang**[1,2] **Ruiyuan Xu**[1,2] **Jin Qin**[1,2] **Jiaming Guo**[1] **Chenxi Wang**[1] **Ling Li**[2,4]
**Qi Guo**[1] **Yunji Chen**[1,2]*

[1] State Key Lab of Processors, Institute of Computing Technology, CAS
[2] University of Chinese Academy of Sciences
[3] Institute of Microelectronics, CAS
[4] Intelligent Software Research Center, Institute of Software, CAS

## Abstract

The rise of GPU-based high-performance computing (HPC) has driven the widespread adoption of parallel programming models such as CUDA. Yet, the inherent complexity of parallel programming creates a demand for the automated sequential-to-parallel approaches. However, data scarcity poses a significant challenge for machine learning-based sequential-to-parallel code translation. Although recent back-translation methods show promise, they still fail to ensure functional equivalence in the translated code. In this paper, we propose **QiMeng-MuPa**, a novel **Mu**tual-Supervised Learning framework for Sequential-to-**Pa**rallel code translation, to address the functional equivalence issue. QiMeng-MuPa consists of two models, a Translator and a Tester. Through an iterative loop consisting of Co-verify and Co-evolve steps, the Translator and the Tester mutually generate data for each other and improve collectively. The Tester generates unit tests to verify and filter functionally equivalent translated code, thereby evolving the Translator, while the Translator generates translated code as augmented input to evolve the Tester. Experimental results demonstrate that QiMeng-MuPa significantly enhances the performance of the base models: when applied to Qwen2.5-Coder, it not only improves Pass@1 by up to 28.91% and boosts Tester performance by 68.90%, but also outperforms the previous state-of-the-art method CodeRosetta by 1.56 and 6.92 in BLEU and CodeBLEU scores, while achieving performance comparable to DeepSeek-R1 and GPT-4.1. Our code is available at `https://github.com/kcxain/mupa`.

## 1 Introduction

With the rapid advancement of high-performance computing (HPC) on GPUs, parallel programming models like CUDA have become a critical tool for implementing computationally intensive tasks. Different from the simplicity and familiarity of sequential programming languages such as C, the parallel programming paradigm of CUDA is fundamentally more complex, requiring expertise in both parallel logic and hardware-specific concepts, including threads, blocks, and memory hierarchy. This complexity makes manual sequential-to-parallel translation and optimization a highly challenging task. Consequently, developing automated methods for sequential-to-parallel translation holds immense potential to accelerate computations, reduce development effort, and fully harness the power of GPU platforms.

---

*Corresponding author.

39th Conference on Neural Information Processing Systems (NeurIPS 2025).

In recent years, machine learning methods have brought tremendous advances to automated sequential-to-parallel translation. Unfortunately, the scarcity of CUDA corpora significantly limits the CUDA programming capabilities of supervised learning methods [2]. To address this issue, unsupervised back-translation method in the field of Neural Machine Translation (NMT) [Sennrich et al., 2016, Edunov et al., 2018, Artetxe et al., 2018] has been widely applied in code translation [Lachaux et al., 2020, TehraniJamsaz et al., 2024]. However, the syntactically and functional correctness of code translation is unable to be ensured by straightforwardly applying back-translation methods. Thus, works like BabelTower [Wen et al., 2022] and MIRACLE [Zhu et al., 2024] use static analysis and incorporate compiling feedback to filter syntactically correct code, but they struggle to guarantee functional equivalence between the original and translated code. Although recent Large Language Models (LLMs) have shown strong potential in general code generation, achieving functional equivalence of translated code remains important for supervised fine-tuning, especially for languages with low data availability such as CUDA. In conclusion, the functional equivalence issue of sequential-to-parallel translation due to data scarcity remains critical but unsolved.

Unit test generation methods such as TransCoder-ST [Rozière et al., 2021] bring new opportunities for generating and filtering functional equivalence data by using Evo-suite [Fraser and Arcuri, 2011]. However, Evo-suite relies on a genetic algorithm which is extremely inefficient, and requires extensive adaptation work for different programming languages. LLMs can also be used to generate unit tests for programs, offering faster speeds, better scalability, and improved readability [Li and Yuan, 2024], but unit test data for supervised fine-tuning are still needed.

To address the functional equivalence issue in sequential-to-parallel translation, we propose a Mutual-Supervised Learning framework, QiMeng-MuPa. The key insight is to establish a closed-loop that simultaneously performs code translation and functional equivalence verification, where the two components mutually generate data to enhance their generation ability and improve their performance collectively. QiMeng-MuPa consists of two models: a Translator and a Tester, both of which are language-agnostic. The Translator converts source code into target code, meanwhile the Tester generates valid unit test for the given source code. To simultaneously enhance the Translator and the Tester, the proposed framework consists of two steps: Co-verify and Co-evolve. In Co-verify step, the unit tests generated by the Tester are used to validate the functional equivalence and filter the code generated by the Translator. In the Co-evolve step, the filtered generated code are used to train both the Translator and the Tester. These two steps are performed iteratively to simultaneously enhance the Translator and the Tester, i.e. the stronger Translator generates more correct translated code as the augmented input for the Tester, while the stronger Tester generates more valid unit tests to filter the translated code for training the Translator. With the improvement of both the Translator and the Tester during the iteration, data scarcity can be addressed by the Translator, and functional equivalence can be guaranteed by the Tester. Thus, filtered by unit tests, QiMeng-MuPa can achieve superior performance compared to non-filtered or only compile-filtered methods, as shown in Figure 4.

We conducted extensive experiments to evaluate the proposed method. More than just evaluating BLEU and CodeBLEU scores, which primarily reflect syntactic equivalence and have little correlation with functional correctness (see Figure 9), we use Pass@k as the metric that directly reflects the functional equivalence of translation. Experimental results demonstrate that QiMeng-MuPa significantly enhances model performance. When applied to Qwen2.5-Coder, it improves the Translator's Pass@1 by up to 28.91% and boosts the Tester's performance by 68.90%; when applied to Llama3, it increases Pass@1 by up to 29.44% and improves Tester performance by 59.53%. In terms of translation quality, QiMeng-MuPa outperforms the previous state-of-the-art method CodeRosetta by 1.56 BLEU and 6.92 CodeBLEU, while achieving Pass@k performance comparable to DeepSeek-R1 and GPT-4.1. Moreover, training Qwen3-0.6B on our filtered and verified data enables it to match the performance of GPT and DeepSeek series. To the best of our knowledge, we are the first to develop a domain-specific LLM capable of automatic code parallelization for HPC.

## 2 Methodology

QiMeng-MuPa involves two models, a Translator and a Tester, which are optimized in an iterative framework. Each iteration consists of two steps: Co-verify and Co-evolve. These two steps alternate

---

[2]There are only 26k repositories on GitHub containing CUDA code, compared to 41M for C.

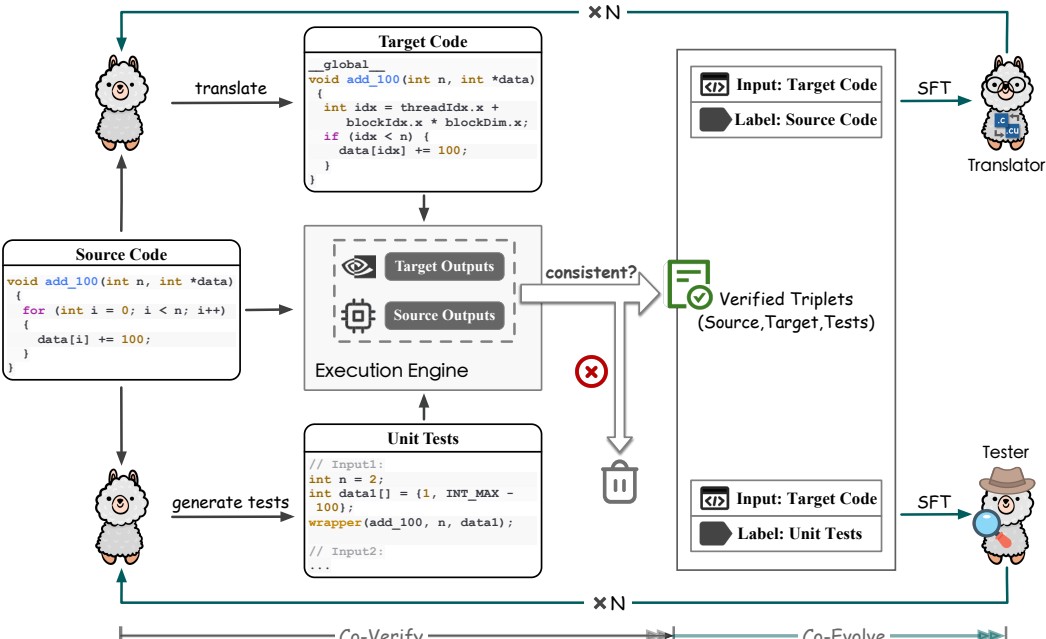

Figure 1: **The Overview of QiMeng-MuPa.** The framework consists of two models: a Translator and a Tester, with two steps: (1) Co-verify: the translated code from the Translator and the corresponding unit tests from the Tester are jointly verified by running on CPU/GPU. If the results of the source and target programs are inconsistent or have compilation/runtime errors, the data is discarded. (2) Co-evolve: the verified parallel (source, target, unit tests) triplets from the Co-verify step are used to fine-tune both the Translator and the Tester via back-translation, improving their performance iteratively until convergence.

until both models converge. We describe the interaction between the two models in these steps in more details below. An overview of the framework is illustrated in Figure 1.

## 2.1 Translator and Tester

Let $\mathbb{L}_s$ and $\mathbb{L}_t$ denote the source set and target set which consist of valid code method strings, respectively. The program $x \in \mathbb{L}_s$ can be viewed as a function. Let $\mathcal{D}(x)$ denote the set of all valid input of program $x$, i.e. $x$ maps a input $t \in \mathcal{D}(x)$ to the outputs $x(t)$. We have two models: the Translator $\mathcal{F}$ and the Tester $\mathcal{G}$.

The Translator $\mathcal{F}$ performs the code translation task, taking input program $x \in \mathbb{L}_s$ and outputting the corresponding program $y \in \mathbb{L}_t$. In this work, we focus on the problem of translating between parallel programs (e.g., CUDA) and sequential programs (e.g., C). Ideally, the translated code $y \in \mathbb{L}_t$ should be semantic equivalent with the program $x \in \mathbb{L}_s$. However, according to Rice's Theorem in computability theory [Rice, 1953], determining whether two programs are semantically equivalent is an undecidable problem. Considering that in the practice, the main purpose of code translation is to generate code which can achieve the equivalent function, we determine to guarantee the functional equivalence of $x$ and $y$. Let $\rightleftharpoons$ denote the functional equivalence which means for any input $t \in \mathcal{D}(x)$, the original code $x$ and the translated code $y$ should output the same results, i.e. $x(t) = y(t)$. Then the task of code translation can be defined as:

**Definition 1 (Code Translation)** *Given the valid code $x \in \mathbb{L}_s$ in source set, the goal of code translation is to learn a Translator $\mathcal{F}$ which can generate the functional equivalent code $y \in \mathbb{L}_t$*

$$\mathcal{F} : \mathbb{L}_s \to \mathbb{L}_t, y = \mathcal{F}(x), \quad s.t. \quad x \rightleftharpoons y,$$
$$i.e. \quad \forall t \in \mathcal{D}(x), y(t) = x(t).$$

The Tester $\mathcal{G}$ is designed to verify the functional equivalence of the translated code. Thus, given the original code $x \in \mathbb{L}_s$, the Tester should be able to generate corresponding valid unit test $\mathcal{G}(x)$

which can evaluate the functional equivalence. Thus, the generated unit test $\mathcal{G}(x)$ should meet two requirements. One is the validity of the generated unit test $\mathcal{G}(x)$, i.e. any input $t \in \mathcal{G}(x)$ should be a valid input in $\mathcal{D}(x)$. The other one is that the generated unit test $\mathcal{G}(x)$ is capable of checking the functional equivalence, i.e. given a pair of functional equivalent codes $x$ and $y$, for any $t \in \mathcal{G}(x)$ as the input, the output $y(t)$ of the translated code $y$ should be the same with the output $x(t)$ of the original code $x$. Then the task of unit test generation can be defined as:

**Definition 2 (Unit Test Generation)** *Given a source code $x \in \mathbb{L}_s$ and the corresponding functional equivalent target code $y \in \mathbb{L}_t$ and $x \rightleftharpoons y$, the goal of unit test generation is to learn a Tester $\mathcal{G}$ to generate valid unit test $\mathcal{G}(x)$ which can show the functional equivalence of $x$ and $y$*

$$\forall t \in \mathcal{G}(x), s.t. \quad t \in \mathcal{D}(x) \quad and \quad x(t) = y(t).$$

## 2.2 Mutual-Supervised Learning Framework

Both the Translator and the Tester suffer from a lack of supervised data, rendering traditional supervised learning methods infeasible. Considering the lack of supervised data of code translation, we start from a collection of multilingual program corpora as the source set. This corpora contains a large, diverse set of programs, including both sequential and parallel programs. The corpora are not aligned between the two languages, making it be easily obtainable through automatic methods (e.g., scraping from GitHub). However, since there is no alignment between the two languages, the corpora cannot be applied for supervised learning.

Fortunately, the Translator $\mathcal{F}$ and the Tester $\mathcal{G}$ can mutually generate new data for each other based on the monolingual corpora, addressing the issue of aligning data scarcity. The Tester $\mathcal{G}$ is able to generate unit tests $\mathcal{G}(x)$ to verify the functional equivalence of the generated code $\mathcal{F}(x)$ from the Translator $\mathcal{F}$, so that the functional equivalent code pair $(\mathcal{F}(x), x)$ filtered by the Tester can be used as the supervised training data of the Translator. In turn, the Translator $\mathcal{F}$ is able to generate some functional equivalent code $\mathcal{F}(x)$ whose unit test is the same with the original code $x$, i.e. $\mathcal{G}(x)$, thus, the code pair $(\mathcal{F}(x), \mathcal{G}(x))$ can be used as the supervised training data of the Tester. Furthermore, it is essential to simultaneously improve both the Translator and the Tester. Optimizing only the Translator without enhancing the Tester may lead to the generation of flawed unit tests, potentially misjudging the correctly translated data. Conversely, if the Translator remains unimproved, it will inevitably produce erroneous translation code, hindering the generation of useful unit tests. In conclusion, we can establish a closed-loop based on the Translator and the Tester to mutually generate high-quality data and enhance their generating ability collectively.

Based on the above analysis, we propose the QiMeng-MuPa, which iteratively filters data and simultaneously optimizes both the Translator and the Tester. The Translator and the Tester are initialized with a general LLM which has some code generation capabilities though the capability is limited. Since the corpora is multilingual, containing both sequential and parallel programs, both the Translator and the Tester are language-agnostic. Namely, the Translator is able to perform C-to-CUDA and CUDA-to-C translation simultaneous, while the Tester is able to generate unit tests for both C codes and CUDA codes. The proposed framework iteratively improves both the Translator and the Tester in a closed-loop. Each iteration consists of two steps, Co-verify and Co-evolve.

**Co-verify** In the Co-verify step, the generated code from the Translator and the generated unit test from the Tester generates are verified collectively. In the $i$-th iteration, given the source code $x$, denote $p(y|x; \theta_i)$ as the conditional probability distribution of the target code $y$ generated from the Translator, meanwhile, denote $p(T|x; \phi_i)$ as the conditional probability distribution of the unit tests $T = \{t_1, t_2, ..., t_n\}$ generated from the Tester. We define $x(T) = y(T)$ to mean $\forall t \in T, x(t) = y(t)$.

To collect all the functional equivalent data, we examine all the source code $x \in \mathbb{L}_s$ by evaluating it with the corresponding translated code $y \sim p(y|x; \theta_i)$ and generated unit tests $T \sim p(T|x; \phi_i)$, to evaluate whether $x(T) = y(T)$. We can only evaluating whether $x(T) = y(T)$ based on the actual results of the program's execution on the OS and CPU/GPU. Once $x(T) = y(T)$, it indicates two things: 1) $T \subset \mathcal{D}(x)$, meaning all the unit tests generated by the Tester are valid. 2) The Translator's results exhibit functional equivalence. Otherwise, if $x(T) \neq y(T)$, we cannot determine whether the inequality is due to the translation error of the Translator or the unit test generation error of the Tester. Thus, the correctness of these two model must be evaluated simultaneously. This is why the two models are referred to as Co-verify.

After all data in the source set has been evaluated, we can obtain the $i$-th high-quality supervised training data $\mathcal{S}_i^{\mathbb{L}}$, where each item is a verified triplet:

$$\mathcal{S}_i^{\mathbb{L}} = \{(x, y, T) \mid x \in \mathbb{L}, y \sim p(y|x; \theta_i), T \sim p(T|x; \phi_i), x(T) = y(T)\}$$

**Co-evolve** In the Co-evolve step, verified (source, target, unit tests) triplets from Co-verify step are extracted as the supervised training dataset which is used to fine-tune both the Translator and the Tester. In the $i$-th iteration, giving the $i$-th filtered training data $\mathcal{S}_i^{\mathbb{L}}$, we fine-tune both the Translator and the Tester via back-translation [Lachaux et al., 2020]. For the Translator, we take the target code $y$ generated in the Co-verify step as input, with the original code $x$ serving as the reference, to optimize the Translator with the following cross-entropy loss function:

$$\mathcal{L}_{\text{trans}}(\theta_i) = - \mathbb{E}_{(x,y,T) \sim \mathcal{S}_i^{\mathbb{L}}} \left[ \sum_{j=1}^{|x|} \log p(x_j | x_{<j}, y; \theta_i) \right]$$

For the Tester, we take the target code $y$ generated by the Translator as input and the generated unit tests $t$ for the original code $x$ as output, to optimize the Tester with the following cross-entropy loss function:

$$\mathcal{L}_{\text{test}}(\phi_i) = - \mathbb{E}_{(x,y,T) \sim \mathcal{S}_i^{\mathbb{L}}} \left[ \sum_{j=1}^{|t|} \log p(T_j \mid T_{<j}, y; \phi_i) \right]$$

By optimizing the above loss function, we can evolve the Translator and the Tester.

$$\theta_{i+1} = \arg\min_{\theta} \mathcal{L}_{\text{trans}}(\theta_i) \qquad \phi_{i+1} = \arg\min_{\phi} \mathcal{L}_{\text{test}}(\phi_i)$$

Note that the source set $\mathbb{L}_s$ is initialized with a collection of multilingual program corpora including both C and CUDA programs, meanwhile the Translator and the Tester are both language-agnostic. Therefore, in each Co-verify step, the filtered training dataset $\mathcal{S}_i^{\mathbb{L}}$ also contains both C and CUDA programs. Then, in each Co-evolve step, the Translator is trained with both C-to-CUDA and CUDA-to-C supervised data, meanwhile the Tester is trained with both C to unit tests and CUDA to unit tests supervised data. Thus, the performance of the two models on their corresponding tasks in both languages will improve. After that, in the co-verify step of next iteration, the Translator will generate more functional equivalent translated code, meanwhile the Tester will generate more valid unit tests, thereby filtering out more valid and diverse triplets for $\mathcal{S}_{i+1}^{\mathbb{L}}$. This brings the improvement of both models in the further iterations until convergence. While it is possible to train separate Translators and Testers for C and CUDA, we find that language-agnostic models perform better. The possible reason is that the amount of training data is the most important factor, so combining multiple languages to train language-agnostic models can achieve better results.

## 3 Experimental Setup

### 3.1 Dataset

**Unpaired training set.** We filter the unaligned training set in BabelTower [Wen et al., 2022] which contains 501,732 C functions and 129, 497 CUDA kernel functions. Considering these functions may cannot be executed due to calls to third-party libraries or user-defined functions, we filtered these functions through compilation, obtaining 14,687 valid C functions and 28,756 valid CUDA kernel functions as our training set.

**Paired test sets.** The validation set and test set in BabelTower [Wen et al., 2022] consist of 364 pairs of C and CUDA functions. We use GPT-4 [OpenAI et al., 2024] to generate unit tests for each pair, and ultimately filter out 233 pairs after compilation, each with 5 unit tests. Our generated tests achieve an average line coverage of 99%, demonstrating their robustness in exercising the functionality of the translated programs. An analysis of the test set's coverage and difficulty is included in the appendix A.

Table 1: **Experimental Results on C-to-CUDA Translation.** We evaluate the performance of framework using Llama3, Qwen2.5-Coder, and Qwen3, and compare it with prior state-of-the-art work CodeRosetta as well as strong general-purpose LLMs. Evaluation is conducted across multiple metrics: BLEU, CodeBLEU, Compile Pass (CPass), and Pass@{1,5,10}. (QiMeng-MuPa$^T$ denotes fine-tuning with data filtered by Qwen2.5-Coder.)

| Model Name | Size | Text-Similarity (%) | | Functional-Equivalence (%) | | | |
|---|---|---|---|---|---|---|---|
| | | BLEU | CodeBLEU | CPass | Pass@1 | Pass@5 | Pass@10 |
| GPT-4o | - | 76.28 | 46.56 | 89.66 | 83.69 | 92.50 | 94.26 |
| GPT-4.1 | - | 75.39 | 36.41 | 83.26 | 80.28 | 92.71 | 94.68 |
| DeepSeek-V3 | 671B | 79.66 | 45.05 | 91.85 | 85.61 | 94.87 | 96.40 |
| DeepSeek-R1 | | 71.40 | 39.00 | 87.98 | 82.23 | 95.46 | **97.63** |
| TransCoder | 0.6B | 75.43 | 78.06 | - | - | - | - |
| CodeRosetta | 0.8B | 83.40 | 78.84 | 81.10 | 75.96 | - | - |
| Qwen3 | 0.6B | 61.25 | 36.11 | 31.33 | 11.59 | 20.56 | 22.18 |
| + QiMeng-MuPa$^T$ | | 81.56 | 82.01 | 94.42 | 84.44 | **96.15** | 97.33 |
| Qwen2.5-Coder | 7B | 72.07 | 47.79 | 72.10 | 56.24 | 77.73 | 80.90 |
| + QiMeng-MuPa | | 84.96 | 84.98 | **95.71** | 85.15 | 95.73 | 97.00 |
| Llama3-Instruct | 8B | 69.05 | 42.93 | 68.24 | 56.12 | 71.64 | 74.03 |
| + QiMeng-MuPa | | **86.68** | **86.63** | 92.70 | **85.56** | 93.79 | 94.57 |

**Code Wrappers.** To support unit test execution and obtain results for C and CUDA, we preprocess the code with two types of wrappers: 1) Function Call Wrapper: Some functions modify data via pointer arguments (e.g., output matrices) instead of returning values. Using C++ templates and regex, we generate wrappers that detect parameter types and print modified arrays after execution, capturing both return values and side effects. 2) CUDA Kernel Wrapper: Since CUDA kernels require control flow logic (e.g., memory initialization, data transfer, grid and block allocation) for invocation, we use base model (the prompt is provided in Appendix G) to create a wrapper for each CUDA kernel in the monolingual dataset. The wrapper replicates the kernel's interface, adds control flow logic, and invokes the actual CUDA kernel, enabling CUDA kernels to be called like C functions.

## 3.2 Base model & fine-tuning

We use Llama3-8B-instruct [Grattafiori et al., 2024] and Qwen2.5-Coder-7B [Yang et al., 2024a] as the base model for the Translator, Tester, and CUDA Wrapper. For generation, when generating with the closed-source models, we use a one-shot prompt. For generating with the trained models and during training, we use only task-specific prompts. More details for fine-tuning can be found in Appendix D.2 and two different prompts for both code translation and unit test generation are detailed in Appendix G.

## 3.3 Metrics

We use four key metrics (**BLEU** [Papineni et al., 2002], **CodeBLEU** [Ren et al., 2020], **Compile Pass (CPass)** and **Pass@k** [Chen et al., 2021]) to evaluate Code Translation task and **Valid Input (VT)** to evaluate Unit Test Generation Task (see Appendix D.1 for details).

## 3.4 Baselines

The main baselines we compare to are the following approaches. TransCoder [Lachaux et al., 2020] and CodeRosetta [TehraniJamsaz et al., 2024] adopt encoder-decoder Transformer models for code translation. While CodeRosetta represents the current state-of-the-art, it only reports similarity-based metrics (BLEU and CodeBLEU), which insufficiently capture true translation quality. Besides, we evaluate the powerful closed-source models, all major OpenAI models, including GPT-4o and GPT-4.1 and two leading open-source models: DeepSeek-R1 [DeepSeek-AI et al., 2025] and DeepSeek-V3 [DeepSeek-AI et al., 2024]. DeepSeek-R1 is a reasoning model, achieving state-of-the-

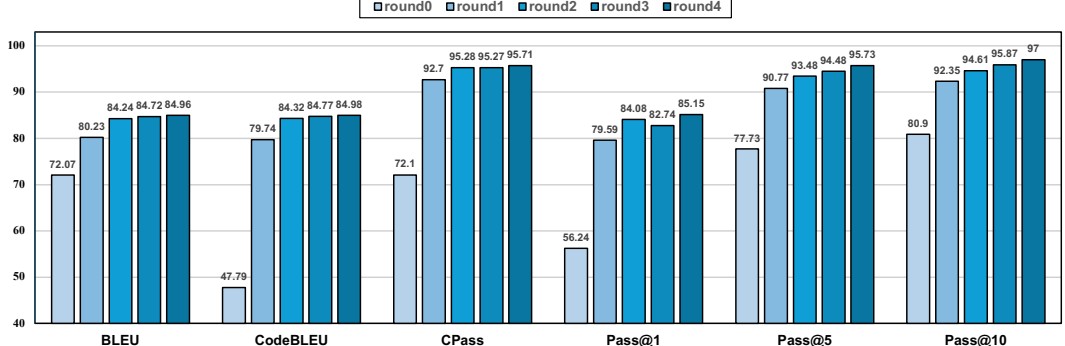

Figure 2: Evaluation of Code Translation across iterations of QiMeng-MuPa based on Qwen2.5-Coder.

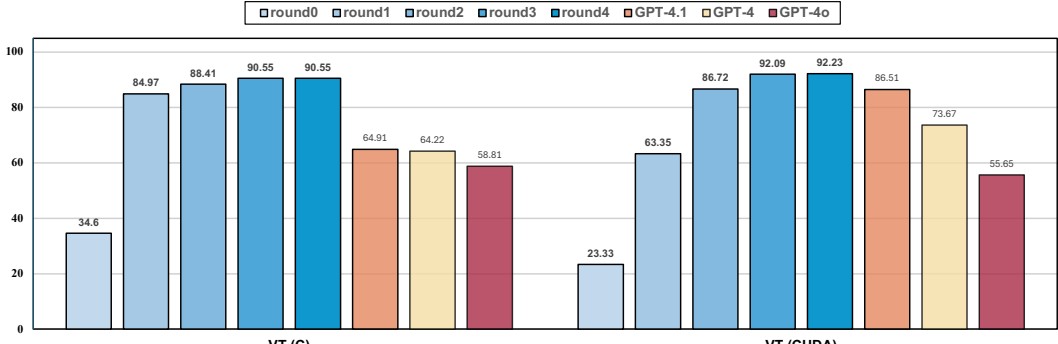

Figure 3: Evaluation of Unit Test Generation Using the VT metric across iterations of QiMeng-MuPa based on Qwen2.5-Coder compared with GPT-4.1, GPT-4o and GPT-4.

art performance on code generation. DeepSeek-V3 is a Mixture-of-Experts (MoE) model optimized for general-purpose tasks.

## 4 Experiments Results

### 4.1 Code Translation

Table 1 and Figure 2 shows the results compared with existing methods. QiMeng-MuPa improves Pass@1 by 28.91% for Qwen2.5-Coder and 29.44% for Llama3. Qwen2.5-Coder enhanced with QiMeng-MuPa surpasses the current state-of-the-art CodeRosetta by 1.56 BLEU and 6.92 CodeBLEU. Regarding Compile Pass and Pass@k that better reflect true model performance, QiMeng-MuPa outperforms the strongest open-source model DeepSeek-R1 by 3.33% on Pass@1, and the leading closed-source model GPT-4.1 by 2.32% on Pass@10. These results demonstrate that the Translator trained with QiMeng-MuPa is effective on sequential-to-parallel code translation, and surpass the current state-of-the-art models on both similarity-based and functional equivalence-based metrics. Figure 12 shows an example where the Translator efficiently and correctly parallelizes the two outer loops, successfully translating their complex logic. In contrast, CodeRosetta produces an incorrect translation. More error analysis and examples can be found in Appendix B and E. We also analyze the performance of CUDA programs generated by the Translator (see Table 5), achieving a maximum speedup of up to 15,259×.

### 4.2 Unit Test Generation

As shown in Figure 3, similar with the Translator, the Tester also achieves state-of-the-art VT metric in generating unit tests for both C and CUDA functions. Among the closed-source or open-source LLM models, the best-performing one is GPT-4.1, but it still 25.64% and 5.72% lower than our Qwen2.5 on the VT metric for C and CUDA, respectively. We analyzed and counted the cases

Table 2: **C and CUDA data filtered in each Co-verify step.** Translator* means freezing the Translator and evolving only the Tester in each iteration of the Co-evolve step. Similarly, Tester* refers to evolving the Translator while freezing the Tester.

| | Code | Iter 1 | Iter 2 | Iter 3 | Iter 4 |
|---|---|---|---|---|---|
| QiMeng-MuPa | C | 759 | 1114 | 1797 | 2079 |
| | CUDA | 2459 | 3947 | 6456 | 8484 |
| Tester* | C | 759 | 1040 | 1205 | 1133 |
| | CUDA | 2459 | 3643 | 3906 | 4676 |
| Translator* | C | 759 | 968 | 1055 | 1315 |
| | CUDA | 2459 | 2783 | 3340 | 4310 |

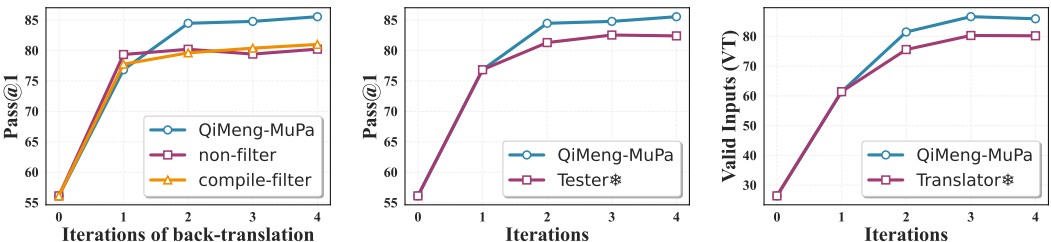

Figure 4: **Ablation Study. Left:** Comparison of the performance of the Translator using Qwen2.5-Coder with non-filter and compile-filter back-translation. **Middle:** Translator performance on Pass@1 when freezing the Tester. **Right:** Tester performance on the VT metric when freezing the Translator.

of invalid single-language unit tests, which mainly include compilation errors and runtime errors. Compilation errors mainly involve issues such as parameter type mismatches, while runtime errors mainly include incorrect understanding of parameter relationships (e.g., segmentation faults due to incorrect array lengths). As shown in Figure 6, the unit tests generated from our method have less compilation errors and runtime errors, and its success rate is larger than that from closed-source or open-source models. Examples of the unit tests generated by the Tester are shown in Appendix F.

## 4.3    Results of iterations

We evaluate each iteration of the Translator and the Tester based on Qwen2.5-Coder. The amount of data filtered by Co-verify in each iteration is shown in Table 2, showing that the filtered paired data for code translation and for unit test generation are both increasing during the iteration. As shown in Figure 2 and Figure 3, the performance of the Translator gradually improves across all four metrics with iterations, saturation at the 4th iteration. Similarly, the Tester's VT metric also gradually improves with each iteration. Particularly, Figure 6 shows from iteration 0 to iteration 1, runtime errors significantly decrease, while compilation errors gradually decrease in the following three iterations. These results demonstrate the effectiveness of the QiMeng-MuPa framework. More valid and paired data of code translation and unit test generation can be filtered in the Co-verify step of each iteration, meanwhile both the Translator and the Tester can be gradually improved by the filtered data in the Co-evolve step of each iteration.

When the proposed method finish, we obtain a verified dataset of 10,563 parallel functions in CUDA and C, along with 5 unit tests per function, automatically constructed through our framework. As shown in Table 1, We trained the Qwen3-0.6B model using this dataset and achieved translation metrics comparable to those of the GPT and DeepSeek series. Moreover, its Pass@1 is 8.48% higher than that of the previous state-of-the-art method at a similar parameter scale, demonstrating the high quality of our filtered dataset.

## 4.4    Ablation Study

**Benefits from Co-verify.**    To evaluate the effectiveness of Co-verify, we compared the proposed method with the performance of 1) non-filtered: the training data was not filtered and back-translated

directly, and 2) compile-filtered: the training data was only filtered by checking compilation. In contrast, QiMeng-MuPa filtered the data with the unit tests to check the functionally equivalence. Figure 4 shows the results of each iteration. Although the two comparison methods initially benefit from a larger amount of training data and show better performance in the early stages, the functionally incorrect data which cannot be filtered eventually hampers their performance. After two iterations, these two methods converge and begin to show a gradual decline in performance, while our approach continues to improve and converges after four iterations of learning. The results demonstrate the effectiveness of our method: Co-verify enhances the quality of the filtered training data, bring more improvement of the performance.

**Benefits from Co-evolve.** To evaluate the mutual contribution of the Translator and the Tester to each other, we freeze one model and train the other. The comparison results in three iterations are shown in Figure 4. Compared to original results of Co-evolve, freezing the other model results in a decrease of 5.72% in Pass@1 and 3.15% in VT for the Translator and Tester. Besides, the amount of data filtered in each iteration is shown in Table 2, showning that the filtered paired data significantly decrease when freezing the Translator or the Tester during the iteration. This suggests that both models contributes to selecting more data in each iteration for training, thereby enhancing the upper bound of model learning. Both the Translator and the Tester are indispensable for QiMeng-MuPa, and train both models simultaneously during the Co-evolve step can achieve the best performance.

# 5 Related Work

**Source-to-Source Translation**. Neural Machine Translation (NMT) has made significant strides [Bahdanau et al., 2014, Sutskever et al., 2014, Vaswani et al., 2017], but translating between programming languages while ensuring functional equivalence remains challenging. Traditional transcompilers (e.g., C2Rust[3], 2to3[4]) rely on expert-defined Abstract Syntax Trees (AST) for source-to-source compilation. Statistical Machine Translation (SMT) methods [Nguyen et al., 2013, 2015] reduce the need for expertise by learning from parallel corpora, but their dependence on large datasets limits adaptation to monolingual corpora. Recent approaches have moved towards unsupervised learning. TransCoder [Lachaux et al., 2020] and TransCoder-IR [Szafraniec et al., 2022] use back-translation to train sequence-to-sequence models on monolingual corpora, avoiding parallel datasets. However, they face challenges in ensuring functional equivalence between corpora, which impacts model performance. BabelTower [Wen et al., 2022] improves translation quality by filtering programs with higher parallelization during back-translation using the ParaBLEU metric, while CodeRosetta [TehraniJamsaz et al., 2024] enhances program understanding by integrating AST Entity Recognition (AER) for sequential-to-parallel translation. To address functional equivalence, TransCoder-ST [Rozière et al., 2021] uses automatically generated unit tests to filter high-quality parallel corpora for iterative fine-tuning with Evo-suite [Fraser and Arcuri, 2011]. In this paper, we propose a novel framework that combines back-translation with unit test models to ensure functional equivalence during training.

**Unit Test Generation**. Unit tests are crucial for verifying program functionality and detecting errors, but manual test writing is labor-intensive and costly, making automated generation [Kumar and Mishra, 2016, Sarsa et al., 2022] a key research focus. Traditional search-based methods [Harman and McMinn, 2010, Fraser and Arcuri, 2011] ensure correctness but struggle with readability and efficiency. Recently, deep learning models, particularly LLMs, have shown promise in unit test generation. Proprietary LLMs excel [Li and Yuan, 2024, Bhatia et al., 2023, Schäfer et al., 2023, Siddiq et al., 2023], but comparative studies [Yang et al., 2024b] show open-source LLMs with supervised fine-tuning often fall short in accuracy and coverage. TestGen-LLM [Alshahwan et al., 2024] improves accuracy with a filtration process without extra training. In this paper, we introduce a novel method for training the Tester. Unlike C++/Python, which benefit from large monolingual datasets, CUDA datasets are limited. We address this by having the Tester first generate tests for C, then using the Translator to convert C to CUDA. This generated data trains the Tester for CUDA tasks, overcoming the monolingual dataset imbalance and achieving excellent results.

---

[3]https://github.com/immunant/c2rust
[4]https://docs.python.org/2/library/2to3.html

# 6 Conclusion

In this paper, we propose a novel framework, QiMeng-MuPa, for sequential-to-parallel code translation. The framework addresses key challenges in the field, including the scarcity of parallel corpora and the need for functional equivalence verification in translation. By integrating two mutually enhancing models, the Translator and the Tester, QiMeng-MuPa iteratively improves both translation accuracy and unit test generation quality. Experimental results demonstrate that QiMeng-MuPa can substantially enhance the performance of base models. Qwen2.5-Coder with QiMeng-MuPa outperforms state-of-the-art methods by 1.56 and 6.92 in BLEU and CodeBLEU, respectively, and both Qwen2.5-Coder and Llama3 achieves Pass@k performance comparable to DeepSeek-R1 and GPT-4.1. Furthermore, our approach helps fill the gap in CUDA dataset availability and contributes a verified dataset of 10,563 parallel functions in CUDA and C, complete with automatically generated unit tests.

## Acknowledgements

This work is partially supported by the NSF of China (Grants No.62525203, U22A2028, 62302483), Strategic Priority Research Program of the Chinese Academy of Sciences (Grants No.XDB0660300, XDB0660301, XDB0660302), CAS Project for Young Scientists in Basic Research (YSBR-029) and Youth Innovation Promotion Association CAS.

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

# A  Analysis of Tests generated for BabelTower

## A.1  Statistics of Test set

We formatted the code using clang-format and used gcov to measure the test coverage for the C code in the test set. We collected statistics on the number of lines per function, number of branches, number of for loops, as well as lines_executed, branches_executed, and taken at least once. The detailed statistics are shown in the table below.

Table 3: Coverage statistics summary.

|  | lines | branches | loops | lines executed (%) | branches executed (%) | taken at least once (%) |
|---|---|---|---|---|---|---|
| **Min** | 4 | 2 | 1 | 83.33 | 50 | 33.33 |
| **Max** | 38 | 24 | 6 | 100 | 100 | 100 |
| **Avg** | 6.79 | 4.47 | 1.68 | 99.17 | 98.91 | 95.8 |

Moreover, only 10 instances of the lines_executed metric fall short of 100%, suggesting that cases where insufficient coverage affects pass@k evaluations are rare and have minimal impact on the results.

The table reveals significant variability in the test set's code length, ranging from a few lines to several dozen. It also includes numerous branches and for loops, making it a robust benchmark for evaluating LLM translation performance.

## A.2  Analysis of the Tester's Coverage During Training

To investigate whether the quality of tests generated by the Tester decreases during training, we analyzed the coverage of the training set in each round using the same method. The results are shown in Figure 5. Since no suitable coverage analysis tool is available for CUDA, we measured the coverage of each CUDA test by applying it to the corresponding translated C version. As shown, the average values of multiple test quality metrics exceed 95%, and only a few kernels with complex branching structures remain difficult to achieve full coverage. As the training rounds progress, both the number and complexity of tested kernels increase, which makes complete coverage more challenging. However, the average coverage remains stable. This observation indicates that functional equivalence is largely preserved and that iterative training does not lead to a degradation in the quality of tests generated by the Tester.

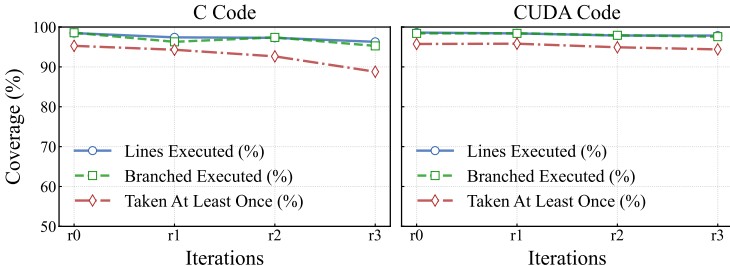

Figure 5: Coverage statistics of training sets across rounds.

## A.3  Observation on the Coverage of Compute-Intensive Code

When generating unit tests for the BabelTower dataset, we also made an interesting observation regarding coverage, as illustrated in Table 3. Unlike the business-oriented Python/Java code, the C/CUDA dataset in BabelTower is primarily composed of computation-intensive kernels, often involving matrix traversal operations. These kernels tend to have few branches but are highly sensitive to numerical values. Consequently, for any given input, the coverage is typically very high, with most values reaching 100%.

# B   Error Type Statistics of Tests

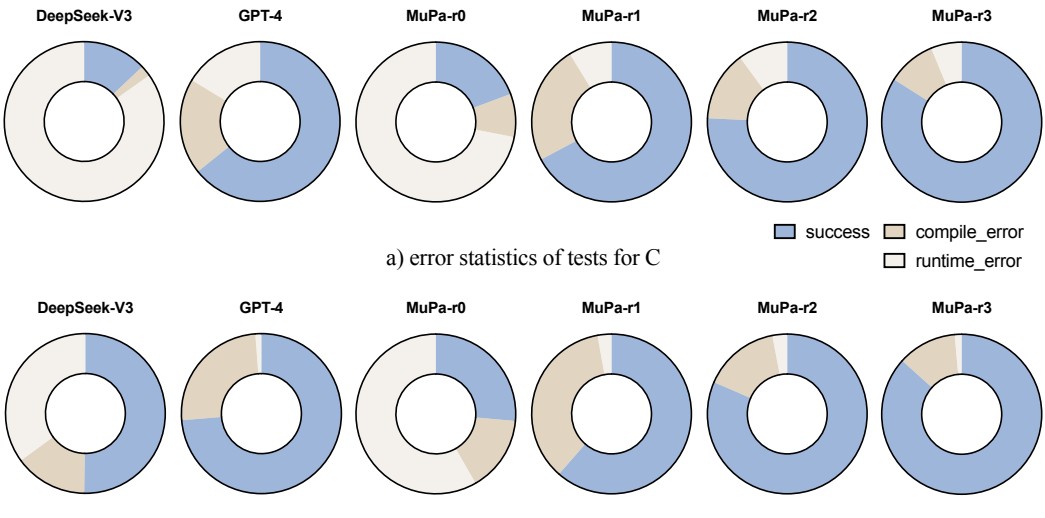

a) error statistics of tests for C

b) error statistics of tests for CUDA

Figure 6: Statistics of error proportions during execution.

We classify the execution errors of co-verify step into 11 types of categories, described as follows:

1. **Type1: Function Overload Resolution.** This typically happens when multiple functions share the same name but differ in parameter lists, and the provided arguments do not clearly match any single overload.

2. **Type2: Insufficient Arguments.** Triggered when a function is called with fewer arguments than required by its definition.

3. **Type3: Argument Type Mismatch:** Happens when the type of an argument passed to a function does not match the type expected by the corresponding parameter in the function's definition.

4. **Type4: Syntax Error: Missing Symbol:** Results from the absence of essential syntax elements such as semicolons (;) or closing braces (}).

5. **Type5: Undefined Identifier.** Occurs when a variable, function, type, or other identifier is used without being declared or defined beforehand.

6. **Type6: Preprocessor Directive Error.** Arises from unexpected tokens or incorrect syntax within #include directives.

7. **Type7: Unrecognized Token:** Triggered by tokens that the compiler does not recognize, often resulting from typos, incorrect syntax, or usage of invalid characters.

8. **Type8: Duplicate Declaration or Standard Library Conflict.** Occurs when a variable or function is declared more than once within the same scope or when a user-defined identifier conflicts with a name in the standard library.

9. **Type9: Logical Sequence, Block Index, or Shared Memory Error.** Pertains to errors in the logical flow of the program, incorrect calculation of block indices in CUDA programming, or improper definition and usage of shared memory.

10. **Type10: Control Flow Error: Variable Initialization Bypass.** Occurs when control flow statements (like goto, break, or return) skip over the initialization of variables, leading to the use of uninitialized or undefined variables.

11. **Type11: CUDA Kernel Call Error.** Happens when there is an attempt to configure a host function (meant to run on the CPU) as a CUDA kernel (meant to run on the GPU).

Table 4: **Error type statistics in the co-verification step.** Statistics are collected from the compilation and execution feedback of 233 parallel data samples in the test set, each with 5 unit tests (total 1165 items). Due to the complexity of feedback, not all error types are listed.

| Model | Type 2 | Type 3 | Type 4 | Type 5 | Type 7 | Type 8 | Type 10 | Type 11 |
|---|---|---|---|---|---|---|---|---|
| GPT-4 | 43 | 1 | 0 | 0 | 2 | 0 | 0 | 0 |
| GPT-4o | 38 | 6 | 35 | 10 | 0 | 0 | 0 | 0 |
| DeepSeek-V3 | 43 | 1 | 19 | 0 | 0 | 0 | 0 | 0 |
| CodeRosetta | 33 | 1 | 22 | 109 | 0 | 0 | 0 | 0 |
| QiMeng-MuPa-r0 | 43 | 1 | 12 | 74 | 0 | 6 | 1 | 10 |
| QiMeng-MuPa-r1 | 44 | 6 | 17 | 29 | 0 | 0 | 0 | 0 |
| QiMeng-MuPa-r2 | 32 | 1 | 0 | 2 | 0 | 0 | 0 | 0 |
| QiMeng-MuPa-r3 | 43 | 1 | 0 | 8 | 0 | 0 | 0 | 0 |

# C   Additional Experiments on QiMeng-MuPa

## C.1   Experimental Results across iterations of QiMeng-MuPa using Llama3

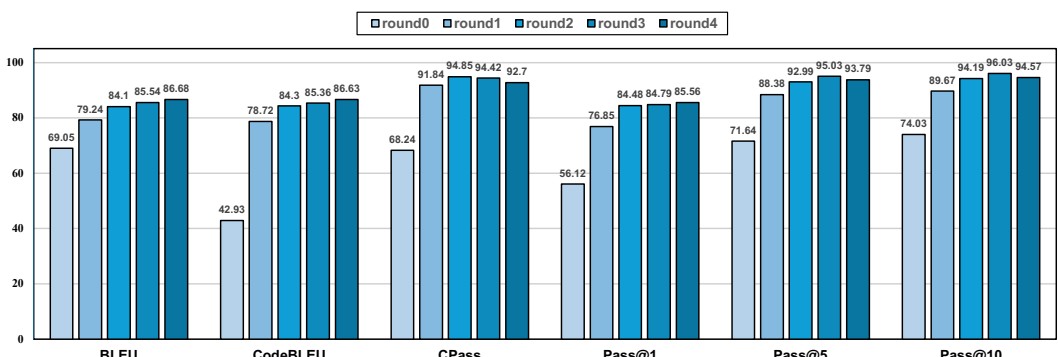

Figure 7: Evaluation of Code Translation across iterations of QiMeng-MuPa based on Llama3.

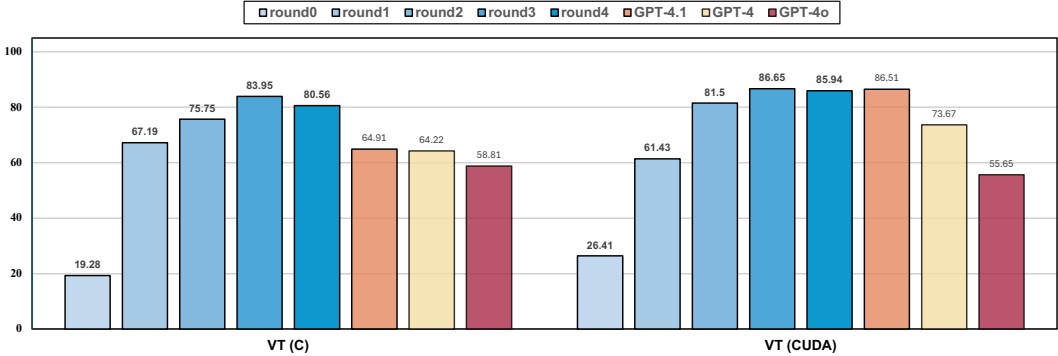

Figure 8: Evaluation of Unit Test Generation Using the VT metric across iterations of QiMeng-MuPa based on Llama3 compared with GPT-4.1, GPT-4o and GPT-4.

## C.2   Analyzing Pearson Correlations Across Evaluation Metrics in Code Translation

We computed the pairwise correlations between BLEU, CodeBLEU, compile pass, and Pass@1, as shown in Figure 9. These results highlight the limitations of using BLEU/CodeBLEU alone for code translation evaluation. While BLEU/CodeBLEU correlate with text-based similarity, they fail to capture the correctness and functionality of the generated code. The weak correlations with compile pass/Pass@1 further indicate that BLEU is unsuitable for evaluating code translation, whereas Pass@1 is essential.

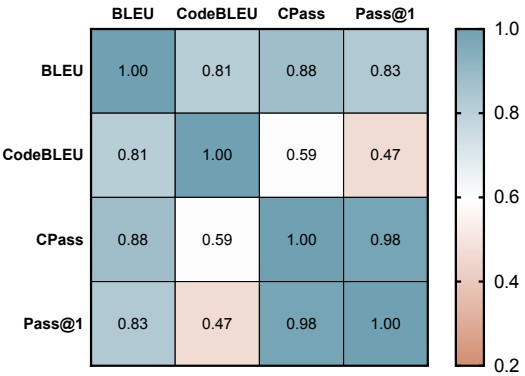

Figure 9: Pearson correlation coefficient of different metrics.

## C.3   Evaluation of CUDA Speedup after Translation

Evaluating the speedup of translated sequential-to-parallel code is essential for assessing performance gains. However, the inputs in the BabelTower and those generated by the Tester are primarily designed to verify functional equivalence, which limits the parallelized version from fully demonstrating its advantages, such as the impact of data loading overhead in CUDA. To overcome this limitation, we used Claude 4-Sonnet to design templates that transform the original inputs into variable-length forms and implemented a random generator to populate them with valid values.

We then systematically evaluated the speedup of models from different training rounds across input sizes ranging from 32 to 16,384. The results, shown in Figure 10, illustrate that later-round models achieve more stable and consistent acceleration as input size increases, indicating improved robustness and scalability in the translated CUDA programs.

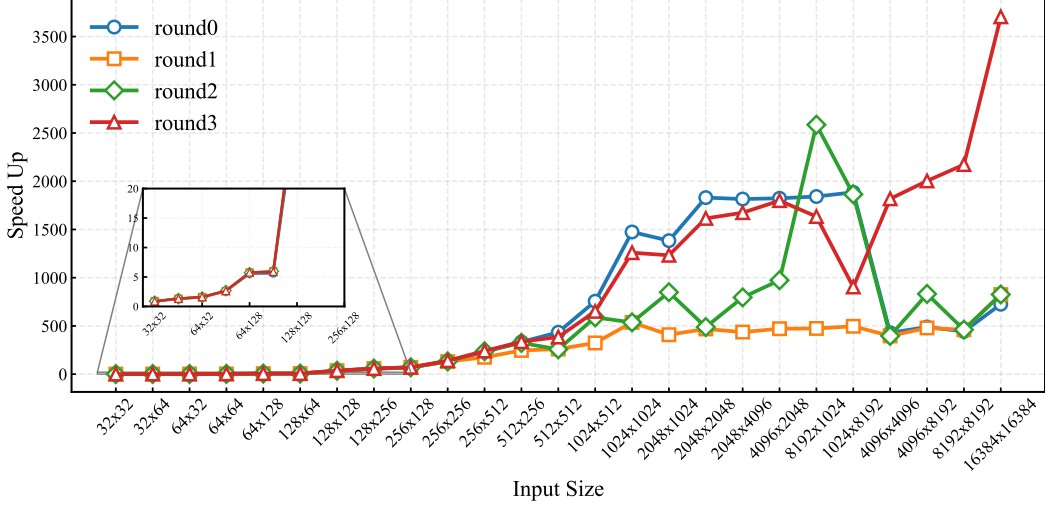

Figure 10: **Speedup results of Tester across training rounds under different input sizes.** Each curve represents a different training round (round0–round3), and the Speedup is measured as the runtime ratio between the translated CUDA code and its sequential counterpart.

We also manually selected three functions from the translation results in test set (shown in Figure 14) and adjusted the input matrix sizes. The results in Table 5 show that the QiMeng-MuPa framework achieved a maximum speedup of 15,259.22 after increasing the input size. Our findings also demonstrate SOTA performance on the Pass@1 metric, proving that our model can translate most C

functions into equivalent CUDA functions with significant speedup, saving developers time in writing CUDA code. This underscores our work's contribution to the HPC field.

Table 5: Evaluation of speedup for cross_correlate, gemm, and transpose using CUDA translated by QiMeng-MuPa-r3 (Llama3-8B).

| kernel | | Matrix Size | | | | | | |
|---|---|---|---|---|---|---|---|---|
| | | 32 | 128 | 512 | 1024 | 2048 | 4096 | 8192 |
| cross_correlate | C (ms) | 0.0022 | 0.0397 | 2.6646 | 11.419 | 85.974 | 506.9 | 2550 |
| | CUDA (ms) | 0.0016 | 0.0017 | 0.0029 | 0.0074 | 0.0254 | 0.4542 | 1.8162 |
| | △Speed up | 1.38↑ | 23.12↑ | 891.65↑ | 1525.5↑ | 3381.3↑ | 1115.8↑ | 1404↑ |
| gemm | C (ms) | 0.0543 | 3.1467 | 198.52 | 1586.4 | 12629 | - | - |
| | CUDA (ms) | 0.0067 | 0.0311 | 0.2199 | 0.4332 | 0.8276 | - | - |
| | △Speed up | 8.01↑ | 101↑ | 902.52↑ | 3661.9↑ | 15259↑ | - | - |
| transpose | C (ms) | 0.0013 | 0.0544 | 0.9922 | 6.5693 | 23.986 | 84.911 | 395.02 |
| | CUDA (ms) | 0.0014 | 0.0014 | 0.0025 | 0.0052 | 0.0148 | 0.0467 | 0.1818 |
| | △Speed up | 0.88↓ | 36.47↑ | 384.94↑ | 1258.9↑ | 1614.4↑ | 1817.6↑ | 2172.2↑ |

# D  Experimental Details

## D.1  Metrics

**Code Translation.**  First, there are two text similarity-based metrics: **BLEU** [Papineni et al., 2002] which is widely used in both natural language translation and programming language translation. **CodeBLEU** [Ren et al., 2020] which not only considers the text-level similarity of weighed n-gram BLEU but also injects code syntax of abstract syntax trees and code semantics of data-flow graph. **Compile Pass (CPass)** which evaluates the syntactical correctness of the generated code by checking whether it can compile. We use five unit tests for each pair in test set to evaluate **Pass@k** [Chen et al., 2021], a metric that calculate the pass rate of translated code passing all the unit tests. Pass@k is the most important metric, as it truly evaluates functional equivalence. Given the test set $\{x_i, T_i\}_{i=1}^m$ contains the source code $x_i$, unit test $T = \{t_j\}_{j=1}^n$, indicator function $\mathbb{I}[\cdot]$ is 1 for true and 0 for false. We define a functional check to determine whether the translated code $\{y_i\}_{i=1}^m$ is correct:

$$\text{Check}(x, y) = \bigwedge_{j}^{n} \mathbb{I}[x_i(t_{ij}) = y_i(t_{ij})]$$

Then, the Pass@k metric estimates the proportion of source code that can be correctly translated at least once in $k$ attempts:

$$\text{Pass@}k = \mathbb{E}_{\text{source}} \left[ 1 - \frac{\binom{n-c}{k}}{\binom{n}{k}} \right]$$

where $n \geq k$ represents the total number of samples for each source code, and $c$ represents the number of samples that pass the functional check.

**Unit Test Generation.**  To ensure the diversity and complexity of the tests generated by the model, we prompt the model to generate five unit tests each time and then run them. We use **Valid Input (VT)** to evaluate whether all five tests successfully run, written as:

$$\text{VT} = \frac{1}{m} \sum_{i}^{m} \bigwedge_{j}^{n} \mathbb{I}[t_{ij} \in \mathcal{D}(x_i)]$$

## D.2  Inference & Fine-tuning

For inference, we use vLLM [Kwon et al., 2023] to perform inference for all open-source models. For Llama3 and Qwen2.5-Coder, we set the temperature to $T = 1.0$ and $top\_p = 1.0$. For Qwen3, we

adopt the non-think mode and follow the official best practices, setting the temperature to $T = 0.7$, $top\_p = 0.8$, and $top\_k = 20$.

For Fine-tuning, we adopt the same hyper-parameters as existing supervised fine-tuning (SFT) methods [Zheng et al., 2024] for most models: learning rate of $1.0 \times 10^{-5}$, cosine learning rate scheduler, warmup ratio of 0.1, and batch size of 32 in total. Additionally, we use the same chat template as in the instruct fine-tuning phase of Llama3 or Qwen2.5.

### D.3 Execution

All executions in our experiments (e.g., Co-verify and evaluation) are conducted on a CPU (Intel i9-14900KF) and a GPU (RTX 4090 with 128 SMs). We use g++ 9.3.0 to compile C programs and nvcc 12.1 for CUDA programs. During execution, we set a maximum timeout of 60 seconds. Any program that exceeds this limit is treated as a runtime error.

# E    Case Study on Code Translation

```
Source C Function

void squareSerial(float *d_in, float *d_out, int N) {
  for (unsigned int i = 0; i < N; ++i) {
    d_out[i] = pow(d_in[i] / (d_in[i] - 2.3), 3);
  }
}
```

```
QiMeng-MuPa ✔

__global__ void squareKernel(float *d_in, float
    *d_out, int N) {
  const unsigned int lid = threadIdx.x;
  const unsigned int gid = blockIdx.x * blockDim.x
      + lid;
  if (gid < N) {
    d_out[gid] = pow(d_in[gid] / (d_in[gid] -
        2.3), 3);
  }
}
```

```
OpenAI-o1 ✘

__global__ void squareSerial(float *d_in, float
    *d_out, int N) {
  int idx = blockIdx.x * blockDim.x + threadIdx.x;
  if (idx < N) {
    d_out[idx] = powf(d_in[idx] / (d_in[idx] -
        2.3f), 3.0f);
  }
}
```

```
QiMeng-MuPa Verify Results

// Input1:
...
// Input2:
float d_in3[] = {2.3f};
float d_out3[1];
wrapper(squareSerial, d_in3, d_out3, 1);
// Output2:
Source_C_Output: "Return value: void Arguments
    after function call: ([ 2.3 ], [ -1.12221e+23
    ], 1)"
CUDA_Output:      "Return value: void Arguments
    after function call: ([ 2.3 ], [ -1.12221e+23
    ], 1)"
...
```

```
OpenAI-o1 Verify Results

// Input1:
...
// Input2:
float d_in3[] = {2.3f};
float d_out3[1];
wrapper(squareSerial, d_in3, d_out3, 1);
// Output2:
Source_C_Output: "Return value: void Arguments
    after function call: ([ 2.3 ], [ -1.12221e+23
    ], 1)"
CUDA_Output:      "Return value: void Arguments
    after function call: ([ 2.3 ], [ inf ], 1)"
...
```

Figure 11: **Case Study 1: Translating a numerical loop to CUDA.** Given a source C function computing a numerical expression in a loop, our method correctly parallelizes the loop and preserves floating-point behavior across edge cases. In contrast, the OpenAI-o1 translation mismanages boundary conditions and yields incorrect outputs (e.g., `inf` instead of the expected finite value) in verification tests.

**Source C Function**

```c
void cpu_cross_correlate(float *Isg, float *Iss, float *sp, float *gp, int npml, int nnz, int nnx) {
  for (int i1 = npml; i1 < nnz - npml; i1++) {
    for (int i2 = npml; i2 < nnx - npml; i2++) {
      int id = i1 + i2 * nnz;
      float ps = sp[id];
      float pg = gp[id];
      Isg[id] += ps * pg;
      Iss[id] += ps * ps;
    }
  }
}
```

**QiMeng-MuPa ✔**

```c
__global__ void cuda_cross_correlate(float *Isg,
    float *Iss, float *sp, float *gp, int npml,
    int nnz, int nnx) {
  int i1 = threadIdx.x + blockDim.x * blockIdx.x;
  int i2 = threadIdx.y + blockDim.y * blockIdx.y;
  int id = i1 + i2 * nnz;
  if (i1 >= npml && i1 < nnz - npml && i2 >= npml
      && i2 < nnx - npml) {
    float ps = sp[id];
    float pg = gp[id];
    Isg[id] += ps * pg;
    Iss[id] += ps * ps;
  }
}
```

**CodeRosetta ✘**

```c
__global__ void gpu_cross_correlate(float *Isg,
    float *Iss, float *sp, float *gp, int npml,
    int nnz, int nnx) {
  int i1 = blockIdx.x * blockDim.x + threadIdx.x;
  int i2 = blockIdx.y * blockDim.y + threadIdx.y;
  if (i1 < nnz - npml && i2 < nnx - npml) {
    int id = i1 + i2 * nnz;
    float ps = sp[id];
    float pg = gp[id];
    Isg[id] += ps * pg;
    Iss[id] += ps * ps;
  }
}
```

**QiMeng-MuPa Verify Results**

```c
// Input1:
const int npml1 = 1;
const int nnz1 = 3;
const int nnx1 = 3;
float Isg1[nnz1 * nnx1] = {0};
float Iss1[nnz1 * nnx1] = {0};
float sp1[nnz1 * nnx1] = {0.5, 0.7, 0.6, 0.8, 1,
    0.9, 0.3, 0.2, 0.4};
float gp1[nnz1 * nnx1] = {1, 0.8, 0.9, 0.7, 1,
    0.6, 0.4, 0.3, 0.5};
wrapper(cpu_cross_correlate, Isg1, Iss1, sp1, gp1,
    npml1, nnz1, nnx1);
// Output1:
Source_C_Output: "Return value: void Arguments
    after function call: ([ 0, 0, 0, 0, 1, 0, 0,
    0, 0 ], [ 0, 0, 0, 0, 1, 0, 0, 0, 0 ], [ 0.5,
    0.7, 0.6, 0.8, 1, 0.9, 0.3, 0.2, 0.4 ], [ 1,
    0.8, 0.9, 0.7, 1, 0.6, 0.4, 0.3, 0.5 ], 1, 3,
    3)"
CUDA_Output:     "Return value: void Arguments
    after function call: ([ 0, 0, 0, 0, 1, 0, 0,
    0, 0 ], [ 0, 0, 0, 0, 1, 0, 0, 0, 0 ], [ 0.5,
    0.7, 0.6, 0.8, 1, 0.9, 0.3, 0.2, 0.4 ], [ 1,
    0.8, 0.9, 0.7, 1, 0.6, 0.4, 0.3, 0.5 ], 1, 3,
    3)"
...
```

**CodeRosetta Verify Results**

```c
// Input1:
const int npml1 = 1;
const int nnz1 = 3;
const int nnx1 = 3;
float Isg1[nnz1 * nnx1] = {0};
float Iss1[nnz1 * nnx1] = {0};
float sp1[nnz1 * nnx1] = {0.5, 0.7, 0.6, 0.8, 1,
    0.9, 0.3, 0.2, 0.4};
float gp1[nnz1 * nnx1] = {1, 0.8, 0.9, 0.7, 1,
    0.6, 0.4, 0.3, 0.5};
wrapper(cpu_cross_correlate, Isg1, Iss1, sp1, gp1,
    npml1, nnz1, nnx1);
// Output1:
Source_C_Output: "Return value: void Arguments
    after function call: ([ 0, 0, 0, 0, 1, 0, 0,
    0, 0 ], [ 0, 0, 0, 0, 1, 0, 0, 0, 0 ], [ 0.5,
    0.7, 0.6, 0.8, 1, 0.9, 0.3, 0.2, 0.4 ], [ 1,
    0.8, 0.9, 0.7, 1, 0.6, 0.4, 0.3, 0.5 ], 1, 3,
    3)"
CUDA_Output:     "Return value: void Arguments
    after function call: ([ 0.5, 0.56, 0, 0.56, 1,
    0, 0, 0, 0 ], [ 0.25, 0.49, 0, 0.64, 1, 0, 0,
    0, 0 ], [ 0.5, 0.7, 0.6, 0.8, 1, 0.9, 0.3,
    0.2, 0.4 ], [ 1, 0.8, 0.9, 0.7, 1, 0.6, 0.4,
    0.3, 0.5 ], 1, 3, 3)"
...
```

Figure 12: **Case Study 2: Boundary-sensitive loop translation.** The source C function applies a nested loop with custom boundaries. Our translation correctly parallelizes both loops and preserves the boundary conditions. In contrast, CodeRosetta omits necessary checks, potentially leading to out-of-bound memory accesses.

**Source C Function**

```c
void boxesScale_cpu(const float *input, float *output, int dims, float scale0, float scale1, float
    scale2, float scale3) {
  for (int tid = 0; tid < dims; tid++) {
    output[tid * 4] = input[tid * 4] / scale0;
    output[tid * 4 + 1] = input[tid * 4 + 1] / scale1;
    output[tid * 4 + 2] = input[tid * 4 + 2] / scale2;
    output[tid * 4 + 3] = input[tid * 4 + 3] / scale3;
  }
}
```

**QiMeng-MuPa ✔**

```c
__global__ void boxesScale_kernel(const float
    *input, float *output, int dims, float scale0,
    float scale1, float scale2, float scale3) {
  int tid = blockIdx.x * blockDim.x + threadIdx.x;
  if (tid >= dims) {
    return;
  }
  output[tid * 4] = input[tid * 4] / scale0;
  output[tid * 4 + 1] = input[tid * 4 + 1] /
    scale1;
  output[tid * 4 + 2] = input[tid * 4 + 2] /
    scale2;
  output[tid * 4 + 3] = input[tid * 4 + 3] /
    scale3;
}
```

**CodeRosetta ✘**

```c
__global__ void boxesScale_gpu(const T *input, T
    *output, int dims, T scale0, T scale1, T
    scale2, T scale3) {
  int tid = blockIdx.x * blockDim.x + threadIdx.x;
  if (tid < dims) {
    output[tid * 4] = input[tid * 4] / scale0;
    output[tid * 4 + 1] = input[tid * 4 + 1] /
      scale1;
    output[tid * 4 + 2] = input[tid * 4 + 2] /
      scale2;
    output[tid * 4 + 3] = input[tid * 4 + 3] /
      scale3;
  }
}
```

**QiMeng-MuPa Verify Results**

```c
// Input1:
float input3[] = {1, 2, 3, 4, 5, 6, 7, 8};
float output3[8];
wrapper(boxesScale_cpu, input3, output3, 2, 1, 2,
    3, 4);
Source_C_Output: "Return value: void Arguments
    after function call: ([ 1, 2, 3, 4, 5, 6, 7, 8
    ], [ 1, 1, 1, 1, 5, 3, 2.33333, 2 ], 2, 1, 2,
    3, 4)"
CUDA_Output:      "Return value: void Arguments
    after function call: ([ 1, 2, 3, 4, 5, 6, 7, 8
    ], [ 1, 1, 1, 1, 5, 3, 2.33333, 2 ], 2, 1, 2,
    3, 4)"
```

**CodeRosetta Verify Results**

```c
// Input1:
float input3[] = {1, 2, 3, 4, 5, 6, 7, 8};
float output3[8];
wrapper(boxesScale_cpu, input3, output3, 2, 1, 2,
    3, 4);
Source_C_Output: "Return value: void Arguments
    after function call: ([ 1, 2, 3, 4, 5, 6, 7, 8
    ], [ 1, 1, 1, 1, 5, 3, 2.33333, 2 ], 2, 1, 2,
    3, 4)"
CUDA_Output:    "Compilation Error: identifier "T"
    is undefined"
```

Figure 13: **Case Study 3: Loop-based tensor scaling with type handling.** The source C function scales each dimension of a 4-element tensor independently. Our method correctly maps the loop to a CUDA kernel and preserves the correct function signature and type. In contrast, CodeRosetta produces a syntactically invalid kernel with an unresolved template type T, leading to a compilation error.

**Source C**

```c
void cpu_cross_correlate(float * Isg, float * Iss,
    float * sp, float * gp, int npml, int nnz, int
    nnx) {
    int i1, i2, id;
    float ps, pg;
    for (i1 = npml; i1 < nnz - npml; i1++) {
        for (i2 = npml; i2 < nnx - npml; i2++) {
            id = i1 + i2 * nnz;
            ps = sp[id];
            pg = gp[id];
            Isg[id] += ps * pg;
            Iss[id] += ps * ps;
        }
    }
}
```

**Translated CUDA**

```c
__global__ void cuda_cross_correlate(float * Isg,
    float * Iss, float * sp, float * gp, int npml,
    int nnz, int nnx) {
    int i1 = threadIdx.x + blockDim.x *
        blockIdx.x;
    int i2 = threadIdx.y + blockDim.y *
        blockIdx.y;
    int id = i1 + i2 * nnz;
    if (i1 >= npml && i1 < nnz - npml && i2 >= npml
        && i2 < nnx - npml) {
        float ps = sp[id];
        float pg = gp[id];
        Isg[id] += ps * pg;
        Iss[id] += ps * ps;
    }
}
```

**Source C**

```c
void device_gemm(double * __restrict__ A, double *
    __restrict__ B, double * __restrict__ C,
    double alpha, double beta, int M, int N, int K,
    bool A_T = false, bool B_T = false) {
    for (int i = 0; i < M; i++) {
        for (int j = 0; j < N; j++) {
            double temp = 0;
            for (int k = 0; k < K; k++) {
                double left = A_T ? A[k + i * K] :
                    A[i + k * M];
                double right = B_T ? B[j + k * N] :
                    B[k + j * K];
                temp += left * right;
            }
            C[i + j * M] = alpha * temp + beta *
                C[i + j * M];
        }
    }
}
```

**Translated CUDA**

```c
__global__ void device_gemm(double * __restrict__
    A, double * __restrict__ B, double *
    __restrict__ C, double alpha, double beta, int
    M, int N, int K, bool A_T = false, bool B_T =
    false) {
    int j = blockIdx.x * blockDim.x + threadIdx.x;
    int i = blockIdx.y * blockDim.y + threadIdx.y;
    if ((i < M) && (j < N)) {
        double temp = 0;
        for (int k = 0; k < K; k++) {
            double left = A_T ? A[k + i * K] : A[i +
                k * M];
            double right = B_T ? B[j + k * N] : B[k
                + j * K];
            temp += left * right;
        }
        C[i + j * M] = alpha * temp + beta * C[i +
            j * M];
    }
}
```

**Source C**

```c
void device_transpose(double * data, double *
    result, int M, int N) {
    int i, j;
    for (i = 0; i < M; i++) {
        for (j = 0; j < N; j++) {
            result[j + i * N] = data[i + j * M];
        }
    }
    return;
}
```

**Translated CUDA**

```c
__global__ void device_transpose(double * data,
    double * result, int M, int N) {
    int i = blockIdx.y * blockDim.y + threadIdx.y;
    int j = blockIdx.x * blockDim.x + threadIdx.x;
    if ((i < M) && (j < N)) {
        result[j + i * N] = data[i + j * M];
    }
    return;
}
```

Figure 14: **Translation examples of three classical matrix operations.** Each row shows a pair of source C code (left) and its corresponding translated CUDA implementation (right), including cross-correlation, general matrix multiplication (GEMM), and matrix transposition. These results are produced by QiMeng-MuPa-r3 (Llama3-8B).

**Source C**

```c
static void simple_dgemm(int n, double alpha,
    const double *A, const double *B, double beta,
    double *C) {
  int i, j, k;
  for (i = 0; i < n; ++i) {
    for (j = 0; j < n; ++j) {
      double prod = 0;
      for (k = 0; k < n; ++k) {
        prod += A[k * n + i] * B[j * n + k];
      }
      C[j * n + i] = alpha * prod + beta * C[j * n
          + i];
    }
  }
}
```

**Translated CUDA**

```c
__global__ void simple_dgemm(int n, double alpha,
    const double *A, const double *B, double beta,
    double *C) {
  int row = blockIdx.y * blockDim.y + threadIdx.y;
  int col = blockIdx.x * blockDim.x + threadIdx.x;
  if (row < n && col < n) {
    double prod = 0;
    for (int k = 0; k < n; ++k) {
      prod += A[k * n + row] * B[col * n + k];
    }
    C[col * n + row] = alpha * prod + beta * C[col
        * n + row];
  }
}
```

**Source C**

```c
void add_bias(float *output, float *biases, int
    batch, int n, int size) {
  int i, j, b;
  for (b = 0; b < batch; ++b) {
    for (i = 0; i < n; ++i) {
      for (j = 0; j < size; ++j) {
        output[(b * n + i) * size + j] +=
            biases[i];
      }
    }
  }
}
```

**Translated CUDA**

```c
__global__ void add_bias_kernel(float *output,
    float *biases, int batch, int n, int size) {
  int index = (blockIdx.x + blockIdx.y *
      gridDim.x) * blockDim.x + threadIdx.x;
  if (index >= n * size * batch)
    return;
  int i = index % size;
  index /= size;
  int j = index % n;
  index /= n;
  int k = index;
  output[(k * n + j) * size + i] += biases[j];
}
```

**Source CUDA**

```c
__global__ void matrixMulCUDA(float *C, float *A,
    float *B, int n) {
  int k;
  int row = threadIdx.y, col = threadIdx.x;
  float sum = 0.0f;
  for (k = 0; k < n; ++k) {
    sum += A[row * n + k] * B[k * n + col];
  }
  C[row * n + col] = sum;
}
```

**Translated C**

```c
void matrixMulCPU(float *C, float *A, float *B,
    int n) {
  for (int i = 0; i < n; ++i)
    for (int j = 0; j < n; ++j) {
      float sum = 0.0f;
      for (int k = 0; k < n; ++k) {
        sum += A[i * n + k] * B[k * n + j];
      }
      C[i * n + j] = sum;
    }
}
```

Figure 15: **Some Cases of Training Set in Back-translation.** This figure demonstrates the translation of various computational functions from C to CUDA using back-translation. The source C code (left) and the translated CUDA kernels (right) are shown for three different operations: matrix multiplication (top), bias addition (middle), and a simple general matrix multiplication (bottom). These cases are part of the training set used to improve the performance and accuracy of the back-translation process.

# F Case Study on Unit Test Generation

To ensure diversity in generation, rather than always selecting the simplest test, we have the Tester generate five tests each time, separated by '//'.

**Source CUDA**

```
// kernel_function
__global__ void pow_kernel(int N, float ALPHA, float * X, int INCX, float * Y, int INCY){
    int i = (blockIdx.x + blockIdx.y * gridDim.x) * blockDim.x + threadIdx.x;
    if(i < N)
        Y[i * INCY] = pow(X[i * INCX], ALPHA);
}

// wrapper_function
void pow_kernel_invoke_in_cpp(int N, float ALPHA, float* X, int INCX, float* Y, int INCY){
    float* d_X;
    float* d_Y;
    cudaMalloc((void**)&d_X, N * sizeof(float));
    cudaMalloc((void**)&d_Y, N * sizeof(float));
    cudaMemcpy(d_X, X, N * sizeof(float), cudaMemcpyHostToDevice);
    int blockSize = 256;
    int numBlocks = (N + blockSize - 1) / blockSize;
    pow_kernel<<<numBlocks, blockSize>>>(N, ALPHA, d_X, INCX, d_Y, INCY);
    cudaMemcpy(Y, d_Y, N * sizeof(float), cudaMemcpyDeviceToHost);
    cudaFree(d_X);
    cudaFree(d_Y);
}
```

**Test 1**

```
int N = 8;
float ALPHA = 2.0;
float X[] = {1.0, 2.0, 3.0, 4.0,
    5.0, 6.0, 7.0, 8.0};
int INCX = 1;
float Y[] = {0.0};
int INCY = 1;
wrapper(
pow_kernel_invoke_in_cpp, N,
    ALPHA, X, INCX, Y, INCY);
```

**Test 2**

```
int N = 5;
float ALPHA = 3.5;
float X[] = {10.0, 20.0, 30.0,
    40.0, 50.0};
int INCX = 1;
float Y[] = {0.0};
int INCY = 1;
wrapper(
pow_kernel_invoke_in_cpp, N,
    ALPHA, X, INCX, Y, INCY);
```

**Test 3**

```
int N = 6;
float ALPHA = 0.5;
float X[] = {-2.0, -1.0, 0.0,
    1.0, 2.0, 3.0};
int INCX = 1;
float Y[] = {0.0};
wrapper(
pow_kernel_invoke_in_cpp, N,
    ALPHA, X, INCX, Y, INCY);
```

**Test 4**

```
int N = 3;
float ALPHA = 1.2;
float X[] = {10.0, 20.0, 30.0};
int INCX = 1;
float Y[] = {0.0};
int INCY = 1;
wrapper(
pow_kernel_cuda_invoke_in_cpp,
    N, ALPHA, X, INCX, Y,
    INCY);
```

**Test 5**

```
int N = 9;
float ALPHA = 0.1;
float X[] = {1.0, 2.0, 3.0, 4.0,
    5.0, 6.0, 7.0, 8.0, 9.0};
int INCX = 1;
float Y[] = {0.0};
int INCY = 1;
wrapper(
pow_kernel_cuda_invoke_in_cpp,
    N, ALPHA, X, INCX, Y,
    INCY);
```

Figure 16: **Case Study 1:** This CUDA kernel performs element-wise power computation. The generated tests vary input values and exponents to capture the kernel's sensitivity to numerical differences.

**Source C**

```c
void get_before_nms_data_cpu(const float *boxes, const float *scores, const int *labels, const int
    *index, float *boxes_out, float *scores_out, int *labels_out, int dims) {
    for (int tid = 0; tid < dims; tid++) {
        if (index[tid] == 0) {
            boxes_out[tid * 4 + 0] = -1;
            boxes_out[tid * 4 + 1] = -1;
            boxes_out[tid * 4 + 2] = -1;
            boxes_out[tid * 4 + 3] = -1;
            scores_out[tid] = -1;
            labels_out[tid] = -1;
        } else {
            boxes_out[tid * 4 + 0] = boxes[tid * 4 + 0];
            boxes_out[tid * 4 + 1] = boxes[tid * 4 + 1];
            boxes_out[tid * 4 + 2] = boxes[tid * 4 + 2];
            boxes_out[tid * 4 + 3] = boxes[tid * 4 + 3];
            scores_out[tid] = scores[tid];
            labels_out[tid] = labels[tid];
        }
    }
}
```

**Test 1**

```c
float boxes1[] = {0, 0, 10, 10};
float scores1[] = {0.9};
int labels1[] = {0};
int index1[] = {0};
float boxes_out1[4];
float scores_out1[1];
int labels_out1[1];
int dims = 1;
wrapper(
get_before_nms_data_cpu, boxes1,
    scores1, labels1, index1,
    boxes_out1, scores_out1,
    labels_out1, dims1);
```

**Test 2**

```c
float boxes2[] = {-1, -1, 3, 3};
float scores2[] = {0.7};
int labels2[] = {1};
int index2[] = {0};
float boxes_out2[4];
float scores_out2[1];
int labels_out2[1];
int dims = 1;
wrapper(
get_before_nms_data_cpu, boxes2,
    scores2, labels2, index2,
    boxes_out2, scores_out2,
    labels_out2, dims2);
```

**Test 3**

```c
float boxes3[] = {1, 2, 3, 4, 5,
    6, 7, 8};
float scores3[] = {0.8, 0.7,
    0.6, 0.5, 0.4, 0.3, 0.2,
    0.1};
int labels3[] = {0, 1, 2, 3, 4,
    5, 6, 7};
int index3[] = {0, 1, 2, 3, 4,
    5, 6, 7};
float boxes_out3[8];
float scores_out3[8];
int labels_out3[8];
int dims = 8;
wrapper(
get_before_nms_data_cpu, boxes3,
    scores3, labels3, index3,
    boxes_out3, scores_out3,
    labels_out3, dims3);
```

**Test 4**

```c
float boxes4[] = {10, 10, 20,
    20};
float scores4[] = {0.9};
int labels4[] = {0};
int index4[] = {0};
float boxes_out4[4];
float scores_out4[1];
int labels_out4[1];
int dims = 1;
wrapper(
get_before_nms_data_cpu, boxes4,
    scores4, labels4, index4,
    boxes_out4, scores_out4,
    labels_out4, dims);
```

**Test 5**

```c
float boxes5[] = {-10, -5, 5,
    10};
float scores5[] = {0.6};
int labels5[] = {1};
int index5[] = {0};
float boxes_out5[4];
float scores_out5[1];
int labels_out5[1];
int dims = 1;
wrapper(
get_before_nms_data_cpu, boxes5,
    scores5, labels5, index5,
    boxes_out5, scores_out5,
    labels_out5, dims);
```

Figure 17: **Case Study 2:** This function filters bounding boxes, scores, and labels based on the index array. The generated tests explore various input scenarios with diverse values and index patterns to reveal the function's sensitivity to numerical conditions.

**Function 1**

```
void dsubtract_matrix(double* a, double* b, double* c, int N){
    for(int idx = 0; idx < N; idx ++){
        c [idx] = a[idx] - b[idx];
    }
}
```

**Test 1 ✘**

```
int data1[] = {0, 0, 0}; // Error Type3: Argument Type Mismatch
int data2[] = {0, 0, 0}; // Error Type3: Argument Type Mismatch
double data3[] = {1.0, 2.0, 3.0};
int N = 3;
wrapper(dsubtract_matrix, data3, data2, data1, N);
```

**Function 2**

```
// kernel_function
__global__ void transposeNaive(int* vector , int* transposed, int size){
    int column = threadIdx.x + blockDim.x * blockIdx.x;
    int row = threadIdx.y + blockDim.x * blockIdx.y;
    if(row < size && column < size)
        transposed[row + column * size] = vector[column + row * size];
}

// wrapper_function
void transposeNaive_cuda_invoke_in_cpp(int* vector, int* transposed, int size){
    int* d_vector;
    int* d_transposed;
    cudaMalloc((void**)&d_vector, size * size * sizeof(int));
    cudaMalloc((void**)&d_transposed, size * size * sizeof(int));
    cudaMemcpy(d_vector, vector, size * size * sizeof(int), cudaMemcpyHostToDevice);
    dim3 numBlocks((size + 15) / 16, (size + 15) / 16);
    dim3 blockSize(16, 16);
    transposeNaive<<<numBlocks, blockSize>>>(d_vector, d_transposed, size);
    cudaMemcpy(transposed, d_transposed, size * size * sizeof(int), cudaMemcpyDeviceToHost);
    cudaFree(d_vector);
    cudaFree(d_transposed);
}
```

**Test 2 ✘**

```
int vector2[] = {1, 2, 3, 4, 5, 6, 7, 8};
int transposed2[4]; // Runtime Error
wrapper(transposeNaive_invoke_in_cpp, vector2, transposed2, 4);
```

Figure 18: Some examples of invalid test cases which cause errors during execution.

# G Prompts

## G.1 CUDA Wrapper

**### User**: Please help me wrap this CUDA kernel to allow user to call it like a C++ function. The wrapper function should:
1. **Keep the input and output parameters and their orders as same the as the original CUDA kernel.**
2. Call the CUDA kernel provided by user inside the wrapper function.
3. The generated code must be in the [CODE] and [/CODE] tags.

Here are examples for you:

CUDA Code:
[CODE]

```cuda
__global__ void add_100_kernel(int numElements, int* data) {
    int idx = blockIdx.x * blockDim.x + threadIdx.x;
    if (idx < numElements) {
        data[idx] += 100;
    }
}
```

[/CODE]

CUDA Code Wrapper:
[CODE]

```cuda
void add_100_cuda_invoke_in_cpp(int numElements, int* data) {
    int* d_data;
    cudaMalloc((void**)&d_data, numElements * sizeof(int));
    cudaMemcpy(d_data, data, numElements * sizeof(int),
        cudaMemcpyHostToDevice);
    add_100_kernel<<<numElements, 1>>>(numElements, d_data);
    cudaMemcpy(data, d_data, numElements * sizeof(int),
        cudaMemcpyDeviceToHost);
    cudaFree(d_data);
}
```

[/CODE]
Your task is to write a wrapper function for the following cuda kernel function, **The generated code must be in the [CODE] and [/CODE] tags**:

CUDA Code:
[CODE]
**{cuda_code}**
[/CODE]

CUDA Code Wrapper:

## G.2 Code Translation

During training or calling the trained model, we use the Input prompt. When calling closed-source models or models not specifically trained for the task, we use Input with one-shot prompt.

---

**### System Prompt**: You are an expert in translating {obj.source} programs to {obj.target} programs. Given the {obj.source} program by User, translate it to {obj.target}. Ensure that the {obj.target} program is exactly the same as the {obj.source} program input and output, and that the semantics of the original code are preserved. Just generate the {obj.target} program and remove any unnecessary comments. Surround the generated {obj.target} program in [{obj.target}] and [/{obj.target}].

**### Input**: {obj.source_code}
**### Input with one-shot (C to CUDA)**:
C Code:

```c
void add_100(int numElements, int *data) {
    for (int idx = 0; idx < numElements; idx++) {
        data[idx] += 100;
    }
}
```

CUDA Code:

```cuda
__global__ void add_100(int numElements, int *data) {
    int idx = blockIdx.x * blockDim.x + threadIdx.x;
    if (idx < numElements) {
        data[idx] += 100;
    }
}
```

Your task is to write a equivalent CUDA kernel function for the following C function:
C Code:
{obj.source_code}
CUDA Code:

---

### G.3 Unit Test generation

During training or calling the trained model, we use the Input prompt. When calling closed-source models or models not specifically trained for the task, we use Input with one-shot prompt.

**### System Prompt**: Your task is to write 5 valid inputs to run the {obj.source} function that performs a specific calculation. You must write the comment "//Input case n:" on a separate line directly above, where n represents the input case number, starting from 1 and increasing by one for each subsequent input case.

**### Input**: {obj.source_code}
**### Input with one-shot (for C)**:
Code:

```c
void add_100(int numElements, int *data) {
    for (int idx = 0; idx < numElements; idx++) {
        data[idx] += 100;
    }
}
```

[INPUTS]

```c
//Input case 1:
int data1[] = {{0}};
add_100(1, data1);

//Input case 2:
int data2[] = {{-100}};
add_100(1, data2);

//Input case 3:
int data3[] = {{1, 2, 3}};
add_100(3, data3);

//Input case 4:
int data4[] = {{INT_MAX - 100}};
add_100(1, data4);

//Input case 5:
int data5[] = {{-50, 0, 50}};
add_100(3, data5);
```

[/INPUTS]
Code: {obj.source_code}

# H    Limitations and Future Work

Although QiMeng-MuPa has demonstrated promising results in sequential-to-parallel code translation, its current development still faces two main limitations: (1) The amount of available training data remains relatively small, constraining the model's ability to fully capture diverse parallel programming patterns. (2) The existing test set focuses mainly on functional correctness and contains relatively simple cases, which may not adequately reflect the model's true potential in handling complex real-world parallel tasks.

In future work, we plan to explore reinforcement learning techniques to enable LLMs to generate more efficient and high-performance parallel kernels, thereby further advancing the automation of HPC code generation.

