# OpenReview forum: "QiMeng-MuPa: Mutual-Supervised Learning for Sequential-to-Parallel Code Translation"
_NeurIPS.cc/2025/Conference — NeurIPS 2025 poster_

### Official Review · Reviewer_YS4J · 2025-07-01

**Clarity:** 2
**Significance:** 2
**Originality:** 2
**Rating:** 4
**Confidence:** 4

**Summary:**

This paper addresses the challenge of ensuring functional equivalence in sequential-to-parallel code translation, where the scarcity of parallelization data makes equivalence guarantees difficult. It proposes a Mutual-Supervised Learning (MSL) framework that enhances equivalence verification and model refinement by establishing an iterative closed loop between a Translator and a Tester.

**Questions:**

Overall, I don’t have any particular suggestions. I find the paper solid if somewhat conventional—its innovation is moderate, but it addresses a very practical problem. I believe it’s necessary for someone to apply a general method to a real-world domain. Therefore, I give it a weak accept.

**Ethical Concerns:**

["NO or VERY MINOR ethics concerns only"]

**Final Justification:**

I remain positive about the paper.

**Limitations:**

yes

**Quality:**

3

**Strengths And Weaknesses:**

Automatically translating existing sequential (CPU) code into efficient parallel (GPU/CUDA) code—while still guaranteeing functional equivalence in the face of scarce data—is indeed a problem worth solving. The authors’ development of a data engine combined with joint supervised fine-tuning to tackle this practical challenge represents a meaningful contribution.

The main concern is that the loop of using filtered data to train a model—and then using the improved model to further filter data—has already been explored in many previous works, so it isn’t particularly novel.

---

> ### Author Rebuttal · Authors · 2025-07-31
>
> We sincerely thank you for your encouraging evaluation and support of our work. We are eager to clarify several aspects of its novelty and look forward to engaging in further constructive discussion!
>
> - In the C to CUDA translation domain, prior SOTA methods like BabelTower[1] and CodeRosetta[2] also employ iterative back-translation, which fundamentally assumes that training progressively reduces noise and improves data quality. However, these methods lack an intermediate filtering step, making it difficult to ensure the quality of the fine-tuning data. To address this issue, MIRACLE[3] uses static analysis and incorporates compiling feedback to filter syntactically correct code, while Transcoder-ST[4] generates unit tests using Evosuite to ensure data quality. However, the former does not truly guarantee functional equivalence, while the latter's unit test generation based on genetic algorithms has significant limitations when applied to C/CUDA. For C/CUDA, EvoSuite's branch coverage-based search method fails to generate numerical edge cases. Our method is significantly more advanced by introducing an additional tester that generates and we innovatively propose the Co-evolve Tester approach to address the issue of generating invalid tests. Our Tester is able to generate a majority of tests with 100% coverage. Moreover, since it is based on a LLM, it captures the program's semantics and produces numerical edge cases, which effectively ensures functional equivalence and thus improves training performance.
> - In instruction alignment, methods like self-alignment[5] filter data by having the model score its own outputs, but this approach still relies on uncertain generations and does not strictly ensure the quality of the generated data. Our work introduces a Tester that executes the translated code to obtain concrete feedback by running it on the CPU/GPU, ensuring the feedback comes from the real physical world.
>
> [1] BabelTower: Learning to Auto-parallelized Program Translation
>
> [2] CodeRosetta: Pushing the Boundaries of Unsupervised Code Translation for Parallel Programming
>
> [3] Semi-supervised code translation overcoming the scarcity of parallel code data
>
> [4] Leveraging Automated Unit Tests for Unsupervised Code Translation
>
> [5] Self-Alignment with Instruction Backtranslation

---

> > ### Comment · Reviewer_YS4J · 2025-08-01
> >
> > Thank you for addressing my questions; I remain positive about the paper.

---

> > > ### Author Response · Authors · 2025-08-05
> > >
> > > We sincerely thank you for your positive evaluation of our paper and for taking the time to carefully review it. We will further refine and incorporate the innovative comparisons into the final version. Thank you again for your support.

---

### Official Review · Reviewer_Ekfk · 2025-07-01

**Clarity:** 3
**Significance:** 3
**Originality:** 3
**Rating:** 4
**Confidence:** 4

**Summary:**

This paper introduces Mutual-Supervised Learning (MuSL), a framework for sequential to parallel code translation, specifically targeting C-to-CUDA generation. The core contribution is a self-supervised, iterative training loop designed to overcome the scarcity of parallel data while enforcing functional correctness. The framework employs two models: a Translator that proposes a parallel code version from a sequential source and a Tester that generates unit tests to verify the functional equivalence of the Translator's output. The process alternates between a "Co-verify" step, where the Tester validates the translation, and a "Co-evolve" step, where code pairs that pass verification are used as high-quality training data to fine-tune both the Translator and the Tester. The authors report that this mutual supervision loop significantly improves the pass rate of translations compared to strong existing baselines, addressing  the need for semantic correctness in automated code generation.

**Questions:**

- Can you provide a quantitative analysis of the quality and coverage (e.g., line, branch, mutation score) of the tests generated by the Tester? A positive response here, demonstrating that the Tester generates non-trivial and comprehensive tests, would significantly increase my confidence in the framework's core verification step. It might be worth stating it in the main text.

- Could you please provide a more precise explanation for the Tester's improvement signal? How does fine-tuning on correct (source, translation) pairs improve its ability to generate more discerning tests for new functions, rather than simply learning to generate tests for a wider variety of functions? A convincing answer is critical for the "co-evolution" claim. In short, why this helps generate better tests?

- Could you provide a systematic analysis of the execution-time speedup of the generated CUDA code versus the sequential C originals across your entire benchmark? A positive result showing that correctness does not come at the cost of performance would substantially strengthen the paper's practical significance.

- What is the wall-clock time and total computational resource requirement for a full MuSL run on your benchmark, and how does this scale with the number of programs and iterations? This is essential for understanding the method's practicality.

- Could you justify the omission of a direct comparison to TransCoder-ST? My evaluation would improve if you could demonstrate the superiority of your learned Tester against a strong, off-the-shelf test generation engine like EvoSuite used in that work.

**Ethical Concerns:**

["NO or VERY MINOR ethics concerns only"]

**Final Justification:**

The original reviewer questions received attention to a certain extent. I have increased my score.

**Limitations:**

While they briefly mention expanding capabilities in the appendix, they fail to discuss the most critical limitations of their work in the main paper. Constructive suggestions for improvement include:

- A dedicated discussion on the fundamental gap between passing unit tests and guaranteeing true functional equivalence. The authors should be upfront that their method is a powerful heuristic for improving confidence in correctness, not a formal guarantee.

- An analysis of the potential failure modes, such as the Tester generating weak or trivial tests, which could lead to a polluted training set of "false positives."

- A transparent discussion of the significant computational cost and scalability challenges, which are a major barrier to the practical application of this method.

- The authors are not fully transparent about the limitations of their work. While they acknowledge some future directions, they do not grapple with the fundamental gap between test-passing and correctness, nor the potential for the Tester to generate trivial or non-discriminating tests.

- Without a clear path to ensuring more robust correctness checks and managing the prohibitive computational cost, it is unlikely that practitioners or researchers will be able to build directly upon this method in its current form.

**Quality:**

3

**Strengths And Weaknesses:**

Strengths:

- The work addresses a problem of high significance for the code translation community.

- The core concept of an iterative, self-supervised loop for data generation and verification is theoretically elegant.

- The work demonstrates good originality in its specific formulation and application.

- The paper is generally well-written and organized, with the core MuSL framework being presented in an intuitive manner. The diagrams and high-level descriptions effectively convey the workflow.

- The reported empirical results, showing a substantial jump in pass rates, are promising and demonstrate the potential of this direction.

-  While using LLMs for code translation and test generation are individually established ideas, the proposed mutual-supervision framework, where the two models iteratively improve each other, is a novel and clever combination of existing techniques. The self-supervised approach to generating a verified, parallel dataset is a creative solution to the data scarcity problem.


Weaknesses:


- In my opinion, the paper's central claim of ensuring "functional equivalence" is not technically sound. It rests on the premise that passing a finite set of generated unit tests is a sufficient proxy for true semantic equivalence, which is undecidable by Rice's Theorem. This is the paper's most significant conceptual weakness. The framework's success is entirely contingent on the Tester's ability to generate comprehensive tests, a capability that is assumed rather than proven.

- Key analyses are missing, rendering the evaluation incomplete. First, there is no systematic evaluation of the performance of the generated parallel code, a critical success metric for any parallelization task. Second, the computational cost and scalability of this iterative compile-and-run loop are not analyzed, which is a major practical concern.

- The paper states the Tester is "enhanced" by fine-tuning on verified code pairs, but it fails to explain how observing a correct (source, parallel_code) pair provides a direct learning signal to generate better, more challenging tests for a new, unseen source program. This feedback loop is crucial to the "co-evolution" claim, and its underspecified nature weakens the paper's argument.

- The originality is slightly diminished by the omission of a direct comparison to highly relevant prior art like TransCoder-ST, which also used a test-based verification step (albeit with a non-learned test generator). Such a comparison would have better isolated the novel contribution of the learned Tester.

---

> ### Author Rebuttal · Authors · 2025-07-31
>
> We sincerely thank you for your thorough analysis and valuable suggestions. Many of your constructive comments are helpful for improving the paper. We address each of them below and welcome further discussion.
>
> > Q1. quantitative analysis of the quality and coverage
>
> A1. We report the average coverage of tests generated by the Tester at each co-verify stage using `gcov`. Since we could not find a suitable coverage analysis tool for CUDA, we report the coverage of each CUDA test by applying it to the translated C version.
>
> |          | lines executed (%) | branched executed (%) | taken at least once (%) |
> | -------- | ------------------ | --------------------- | ----------------------- |
> | r0: C    | 98.48              | 98.57                 | 95.28                   |
> | r0: CUDA | 98.61              | 98.41                 | 95.75                   |
> | r1: C    | 97.41              | 96.29                 | 94.30                   |
> | r1: CUDA | 98.40              | 98.42                 | 95.82                   |
> | r2: C    | 97.29              | 97.40                 | 92.64                   |
> | r2: CUDA | 97.87              | 97.93                 | 94.92                   |
> | r3: C    | 96.29              | 95.27                 | 88.79                   |
> | r3: CUDA | 97.86              | 97.58                 | 94.39                   |
>
> As shown, average values for multiple test quality metrics exceed 95%. Only a few kernels with complex branching remain hard to fully cover. As rounds progress, the number and complexity of tested kernels increase, making complete coverage harder, but average coverage remains stable. This indicates that functional equivalence is largely preserved, and iterative training does not degrade the tester's generated test quality.
>
> For example, even for complex functions like `GCOV` with multiple branches, our Tester achieves 100% coverage.
> ```
> void GOL(int dim, int * grid, int * newGrid) {
>     int iy, ix, id, numNeighbors;
>     int cell;
>     for (iy = 1; iy <= dim; iy++) {
>         for (ix = 1; ix <= dim; ix++) {
>             id = iy * (dim + 2) + ix;
>             numNeighbors = ...
>             cell = grid[id];
>             if (cell == 1 && numNeighbors < 2) newGrid[id] = 0;
>             else if (cell == 1 && (numNeighbors == 2 || numNeighbors == 3)) newGrid[id] = 1;
>             else if (cell == 1 && numNeighbors > 3) newGrid[id] = 0;
>             else if (cell == 0 && numNeighbors == 3) newGrid[id] = 1;
>             else newGrid[id] = cell;
>         }
>     }
> }
> ```
>
> > Q2&W3. provide a more precise explanation for the Tester's improvement signal?  In short, why this helps generate better tests?
>
> A2. Our co-evolve process focuses on improving the tester's ability to generate valid tests, not directly on increasing coverage. Early experiments (e.g., results of round1) show that, since C/CUDA code is compute-intensive and often operates on entire matrices, it involves complex data flows. The model struggles to fully grasp data types and dependencies, leading to invalid tests or runtime errors.
>
> For example, in the `transposeNaive` kernel, the length of `transposed` should match that of `vector`:
>
> but tester generates:
>
> ```
> int vector2[] = {1, 2, 3, 4, 5, 6, 7, 8};
> int transposed2[4]; // Runtime Error
> ```
>
> This indicates that the tester's ability to recognize data dependencies is insufficient.
>
> Therefore, we designed the co-evolve framework: when the tester generates valid tests for the source, we use the target code as input to train the tester to generate valid tests for the target. Through iterative improvement, as shown in Appendix B, this approach significantly reduces runtime errors in generated tests. This indicates that the model is learning to recognize data types and dependencies, thereby enhancing its capabilities.
>
> During training, the tester generates 5 tests at once instead of producing 1 test sampled 5 times. This prevents repeated training on simple tests. Coverage statistics in A1 validate the effectiveness of this strategy, showing that as the model's ability to generate valid tests improves, test quality remains stable.
>
> > Q3&W2. provide a systematic analysis of the execution-time speedup of the generated CUDA code versus the sequential C originals across your entire benchmark.
>
> A3. During translator training from C to CUDA, the labels are always real CUDA kernels collected and cleaned from actual projects by BabelTower. Therefore, from an algorithmic standpoint, it is impossible for the model to sacrifice performance in order to improve accuracy. For a compute-intensive C program, as long as its loops are correctly translated into CUDA thread-level parallelism, there will inevitably be a speedup when the input matrix is sufficiently large.
>
> Since our Tester verifies functional correctness, the generated test cases usually involve small matrices, where kernel launch overhead dominates and masks any speedup. Existing tools cannot generate large-scale inputs for CUDA kernels. To overcome this, we used Claude4-Sonnet to design templates that convert fixed-length inputs into variable-length ones, and implemented a random generator to fill them with valid values. We then systematically evaluated speedup across input sizes from 2048 to 65,536.
>
> The speedup values correspond to the average speedup observed over all successful test cases.
>
> |        | 2048  | 16384 | 65536 |
> | ------ | ----- | ----- | ----- |
> | round0 | 3.459 | 29.39 | 69.59 |
> | round1 | 3.170 | 30.83 | 84.41 |
> | round2 | 3.681 | 33.37 | 120.2 |
> | round3 | 4.189 | 29.03 | 81.98 |
>
> These results show that our translated code consistently achieves significant speedups. The performance improves with larger vector sizes, which aligns well with practical application scenarios. Moreover, as training progresses and accuracy improves across iterations, performance remains stable without any degradation.
>
> > Q5&W4: justify the omission of a direct comparison to TransCoder-ST
>
> A4. EvoSuite is based on genetic algorithms and specifically designed for Java, which is typically used for business logic, where good tests aim to cover all control-flow branches. In contrast, C/CUDA is used for compute-intensive, numerically sensitive code involving batch operations over matrices. For compute-intensive code, good tests should involve target numerical edge cases, which is something that EvoSuite's branch coverage-based search method fails to accomplish. In contrast, in our method, the LLM-based Tester can generate tests that are aware of function semantics and data dependencies based on the inherint code generation ability of LLM, enabling more targeted and meaningful test cases, which EvoSuite often fails to capture.
>
> We show an example demonstrating this difference.
>
> For a CUDA kernel:
>
> ```
> __global__ void Int32ToFloat32(const int32_t * input, float * output, int number) {
>     int i = blockIdx.x * blockDim.x + threadIdx.x;
>     if (i < number) {
>         output[i] = (float) input[i];
>     }
> }
> ```
>
> Out Tester can generate tests like this: `int32_t input5[] = {-2147483648, 0, 2147483647};` to ensure consistent handling of boundary values before and after translation.
>
> In contrast, after translating it to Java, EvoSuite generated the following tests:
>
> ```
> int[] intArray0 = new int[2];
> float[] floatArray0 = new float[0];
> Int32ToFloat32.int32ToFloat32(intArray0, floatArray0, (-371));
>
> int[] intArray0 = new int[2];
> float[] floatArray0 = new float[5];
> Int32ToFloat32.int32ToFloat32(intArray0, floatArray0, 0);
>
> int[] intArray0 = new int[4];
> float[] floatArray0 = new float[3];
> try {
>   Int32ToFloat32.int32ToFloat32(intArray0, floatArray0, 1467);
> } catch(ArrayIndexOutOfBoundsException e) {
>    verifyException("Int32ToFloat32", e);
> }
> ```
>
> These tests only ensure code coverage, but the numerical values are overly simple and clearly inferior to ours.
>
> Additionally, the example on page 18 of the paper further demonstrates that our Tester is capable of generating targeted numerical edge cases. The Tester generates the input `2.3f` based on the denominator value, which led the OpenAI-o1 model to produce `inf` after translation, successfully identifying a translation error.
>
> > W1. the paper's central claim of ensuring "functional equivalence" is not technically sound
>
> A5. First, for C/CUDA code, achieving high coverage is not particularly challenging, as it is typically compute-intensive rather than control-flow intensive. As shown in Q1, our evaluation of coverage demonstrates that the unit tests generated by our Tester consistently achieve high coverage. Second, while neural networks can tolerate a certain amount of data noise such as test-passing but functionally incorrect translations the high coverage of our tests ensures that such noisy samples are limited and do not negatively affect training.
>
> Furthermore, our model surpasses previous SOTA on the test set and performs on par with strong closed-source models after iterative evolution, demonstrating the effectiveness of our method. In addition, the appendix includes numerous manually analyzed cases where our model produces correct translations while others fail, further validating the method's effectiveness.
>
> > Q4. computational cost
>
> A6. We believe computational cost should be considered separately for training and inference. Since in real-world high-performance computing applications training is done only once, inference speed becomes the primary concern.
>
> Our framework involves iterative training with two models, which indeed significantly increases training cost. Our training time cost can be found in reviewer bgUV's A3.
>
> However, when training the Qwen3-0.6B model using our final filtered dataset, we achieve performance far exceeding CodeRosetta at comparable parameter scale. For the 0.6B model, a minimum of 2GB GPU memory is sufficient with decent output speed. Therefore, the computational cost of our proposed method is fully acceptable.

---

> > ### Comment · Reviewer_Ekfk · 2025-08-04
> >
> > The original reviewer questions received attention to a certain extent. I will increase my score.

---

> > > ### Author Response · Authors · 2025-08-05
> > >
> > > Thank you for your positive update in the score and for recognizing the value of our work. We truly appreciate your constructive feedback and thoughtful questions, which have helped us further improve the paper. We will incorporate clarifications addressing your concerns into the final version. Thank you again for your time and support.

---

### Official Review · Reviewer_Rmzg · 2025-07-02

**Clarity:** 3
**Significance:** 3
**Originality:** 2
**Rating:** 4
**Confidence:** 4

**Summary:**

This paper introduces a “Mutual-Supervised Learning” (MSL) framework for translating sequential code (C) to parallel code (CUDA) by pairing a Translator model and a Tester model. The Translator generates candidate code while the Tester produces unit tests, and they iteratively filter and improve each other via a closed-loop process.

**Questions:**

1. Performing equivalence testing.

Can the authors explain how the equivalence testing is performed from the raw inputs? Are the function signatures aligned for the inputs? Are the outputs from program execution serialized before comparison?

2. Vacuous tests discussion.

Can the authors explain how the vacuous tests issue mentioned above is prevented?

**Ethical Concerns:**

["NO or VERY MINOR ethics concerns only"]

**Final Justification:**

Reviewers have incorporated my changes. I maintain my rating.

**Limitations:**

yes

**Paper Formatting Concerns:**

-

**Quality:**

2

**Strengths And Weaknesses:**

Strengths
- Focus on missing functional correctness considerations in prior work on C - CUDA translation
- Adding test cases to BabelTower would be a valuable contribution
- Leveraging back-translation with test-driven
- Extensive experiments across model backbones, metrics, and thorough ablations

Weaknesses
- Vacuous tests. The `Tester` model can cheat and only generate trivial inputs (like all zeros), which might fail to identify bugs during equivalence testing. It is unclear how the approach avoids this failure more.
- Contamination analysis. It is unclear if the monolingual training sets are decontaminated.
- Evaluation suite tests. The test suites only contain 5 test cases which might be less for comlpex C/Cuda functions.

---

> ### Author Rebuttal · Authors · 2025-07-31
>
> Thank you for your positive evaluation and constructive questions. These will help clarify the methodological details in our paper. We respond to each of your points below:
>
> > W1&Q2. Vacuous tests. The `Tester` model can cheat and only generate trivial inputs (like all zeros), which might fail to identify bugs during equivalence testing. It is unclear how the approach avoids this failure more.
>
> A1. Our method avoids generating trivial inputs because it leverages LLMs such as Llama3 and Qwen2.5-Coder, which inherently possess strong capabilities for producing tests targeting compute-intensive code. However, before fine-tuning, these models struggle to handle data types and inter-data dependencies well, resulting in many errors. Therefore, during the Co-Verify stage, instead of having the tester generate one test multiple times, we generate five tests at once and extract regularized results. In the Co-Evolve stage, these five tests are collectively used as SFT outputs for training. This approach avoids training on overly simple tests repeatedly, preventing quality degradation and enhancing the model's ability to generate valid tests.
>
> We report coverage statistics at each co-verify stage, showing that the inputs generated by our method achieve close to 100% coverage and are non-trivial, as noted in Reviewer Ekfk's Q1. Moreover, the learned tester exhibits several desirable properties, i.e. it can generate targeted test cases based on the function's semantics and data characteristics. For example:
>
> ```
> int rows = 1;
> int cols = 1;
> double A[] = {0.0};
> reduce(rows, cols, A);
>
> int rows = 1;
> int cols = 1;
> double A[] = {-100.0};
> reduce(rows, cols, A);
>
> int rows = 2;
> int cols = 2;
> double A[] = {1.0, 2.0, 3.0, 4.0};
> reduce(rows, cols, A);
>
> int rows = 1;
> int cols = 1;
> double A[] = {INT_MAX - 100.0};
> reduce(rows, cols, A);
>
> int rows = 3;
> int cols = 2;
> double A[] = {-50.0, 0.0, 50.0, 0.0, 0.0, 100.0};
> reduce(rows, cols, A);
> ```
>
> The model not only generated matrix `A` with diverse dimensional types, but also initialized it with values such as `INT_MAX - 100.0` to test numerical stability.
>
> > W2. Contamination analysis. It is unclear if the monolingual training sets are decontaminated.
>
> A2. BabelTower performed thorough data cleaning during collection, including removing CUDA code without parallel semantics, eliminating highly duplicated samples, and filtering out code overly similar to the test set. We followed the same strategy and conducted our experiments using the same dataset as BabelTower and CodeRosetta.
>
> > W3. Evaluation suite tests. The test suites only contain 5 test cases which might be less for comlpex C/Cuda functions.
>
> A3. Using too many tests increases output length and execution time, significantly raising experimental cost; too few tests risk failing to ensure functional equivalence. We found that using 5 tests strikes a good trade-off, as shown in the coverage results in Ekfk's Q1 and the training efficiency in bgUV's Q3. Nearly all functions achieved 100% line execution coverage—this is because C/CUDA code is compute-intensive and highly sensitive to input values, so with well-designed test cases, high coverage can be achieved even with a small number of tests.
>
> > Q1. Can the authors explain how the equivalence testing is performed from the raw inputs? Are the function signatures aligned for the inputs? Are the outputs from program execution serialized before comparison?
>
> A4. Since CUDA uses C as its frontend, the two languages share identical data types. Additionally, we enforce that the parameter and return types remain exactly the same across translation, making input sharing straightforward.
>
> Our data analysis shows that most functions involve one/two dimensional arrays. We developed a rule-based C++ template system that automatically detects the parameter and return types. When arrays are present, the system flattens them and prints each element individually.
>
> We compare the function return values along with the flattened parameters as the program's output to verify correctness. For example:
>
> ```
> "cases": {
> "1": "int mat_in1[] = {1, 2, 3, 4};\nint mat_out1[4];\nwrapper(gpu_matrix_transpose, mat_in1, mat_out1, 2, 2);",
> "2": "int mat_in2[] = {1, 2, 3, 4, 5, 6};\nint mat_out2[6];\nwrapper(gpu_matrix_transpose, mat_in2, mat_out2, 2, 3);",
> "3": "int mat_in3[] = {1, 2, 3, 4, 5, 6, 7, 8, 9};\nint mat_out3[9];\nwrapper(gpu_matrix_transpose, mat_in3, mat_out3, 3, 3);",
> "4": "int mat_in4[] = {1, 2, 3, 4, 5, 6, 7, 8, 9, 10, 11, 12};\nint mat_out4[12];\nwrapper(gpu_matrix_transpose, mat_in4, mat_out4, 3, 4);",
> "5": "int mat_in5[] = {0, 0, 0, 0, 0, 0, 0, 0, 0};\nint mat_out5[9];\nwrapper(gpu_matrix_transpose, mat_in5, mat_out5, 3, 3);"
> },
> "outputs": {
> "1": "Return value: void\nArguments after function call: ([ 1, 2, 3, 4 ], [ 1, 3, 2, 4 ], 2, 2)",
> "2": "Return value: void\nArguments after function call: ([ 1, 2, 3, 4, 5, 6 ], [ 1, 4, 2, 5, 3, 6 ], 2, 3)",
> "3": "Return value: void\nArguments after function call: ([ 1, 2, 3, 4, 5, 6, 7, 8, 9 ], [ 1, 4, 7, 2, 5, 8, 3, 6, 9 ], 3, 3)",
> "4": "Return value: void\nArguments after function call: ([ 1, 2, 3, 4, 5, 6, 7, 8, 9, 10, 11, 12 ], [ 1, 5, 9, 2, 6, 10, 3, 7, 11, 4, 8, 12 ], 3, 4)",
> "5": "Return value: void\nArguments after function call: ([ 0, 0, 0, 0, 0, 0, 0, 0, 0 ], [ 0, 0, 0, 0, 0, 0, 0, 0, 0 ], 3, 3)"
> },
> ```

---

> > ### Comment · Reviewer_Rmzg · 2025-08-04
> >
> > Thank you, please add these details to the paper

---

> > > ### Author Response · Authors · 2025-08-05
> > >
> > > We sincerely thank you for taking the time to review our work and for your positive evaluation. Your detailed and thoughtful questions have been very helpful in clarifying important aspects of our paper. We will incorporate these clarifications into the final version. Thank you once again for your support.

---

### Official Review · Reviewer_bgUV · 2025-07-03

**Clarity:** 2
**Significance:** 3
**Originality:** 3
**Rating:** 4
**Confidence:** 3

**Summary:**

This paper proposes a Mutual-Supervised Learning (MuSL) framework for translating sequential code into parallel code. The framework consists of two components, a Translator and a Tester, that iteratively generate data for each other and improve jointly through co-evolution. Experimental results demonstrate the effectiveness of the MuSL framework in enhancing both translation accuracy and functional correctness.

**Questions:**

1. The paper states that "while it is possible to train separate Translators and Testers for C and CUDA, we find that language-agnostic models perform better", and attributes this to the larger amount of training data available when combining multiple languages. Could the authors provide empirical evidence or quantitative results to support this claim? Are there quantitative results comparing language-specific and language-agnostic models? Additionally, if more training data from other languages were included, would the performance improve further?

**Ethical Concerns:**

["NO or VERY MINOR ethics concerns only"]

**Final Justification:**

The clarification provided in the rebuttal largely addresses my concerns, and I will adjust my ratings accordingly.

**Limitations:**

Yes

**Quality:**

3

**Strengths And Weaknesses:**

**Strengths**
+ The topic of code translation, particularly from sequential to parallel code, is important to high-performance computing and software engineering.
+ The proposed Mutual-Supervised Learning framework is language-agnostic, suggesting potential applicability beyond C and CUDA to other programming languages.
+ The experimental evaluation demonstrates the effectiveness of the proposed approach across multiple metrics.

**Weaknesses**
- The framework is claimed to be language-agnostic, but experiments only involve C and CUDA.
- The effectiveness of the proposed method requires further evaluation. It is unclear whether the observed performance improvements are primarily due to the proposed Mutual-Supervised Learning framework, or simply the result of training/fine-tuning the models on the dataset. As shown in Table 1, models with similar parameter sizes such as TransCoder and CodeRosetta achieve comparable results to the proposed approach, raising questions about the contribution of the framework itself versus the impact of supervised training data.
- The evaluation settings should be described in more detail, including specifics such as the hardware configurations. Additionally, the training overhead introduced by the proposed method, especially due to the iterative co-verification and co-evolution steps, should be reported and discussed to assess its practicality and scalability.

---

> ### Author Rebuttal · Authors · 2025-07-31
>
> Thank you for your constructive comments! We address each of the identified weaknesses and questions below.
>
> > W1. The framework is claimed to be language-agnostic, but experiments only involve C and CUDA.
>
> A1. Our work is motivated by the task of Sequential-to-Parallel Code Translation, and our method is specifically designed to address the key challenges in this setting, including three major ones.
>
> 1. LLMs often lack proficiency in the source language (e.g., CUDA).
> 2. Parallel data is scarce, making supervised learning difficult.
> 3. The source and target languages follow different programming paradigms, increasing translation complexity.
>
> In addition, language-agnostic means that our Translator and Tester are jointly trained for bidirectional translation, achieving better performance than training them in a single direction (see the results in A4).
>
> While our method can also be applied to translations between languages like Python and Java, LLMs already demonstrate strong capabilities in these languages, making alignment easier and parallel data more accessible. Therefore, we chose the more challenging C to CUDA setting. Nonetheless, we appreciate your suggestion and plan to explore C to Verilog translation in future work, where similar challenges persist.
>
> > W2. The effectiveness of the proposed method requires further evaluation. It is unclear whether the observed performance improvements are primarily due to the proposed Mutual-Supervised Learning framework, or simply the result of training/fine-tuning the models on the dataset. As shown in Table 1, models with similar parameter sizes such as TransCoder and CodeRosetta achieve comparable results to the proposed approach, raising questions about the contribution of the framework itself versus the impact of supervised training data.
>
> A2. The original training data only consists of monolingual corpora and cannot be directly used for fine-tuning. Our method introduces a framework to construct high-quality parallel data from monolingual sources, which can be used to fine-tuning. Without our framework, directly fine-tunining with the parallel data from back-translation on monolingual data yields worse performance than MuSL. As shown in our ablation study, this leds to a 5.31% perfomace drop on pass@1. Under comparable model sizes and identical data sources, our method outperforms CoderRosetta's CPass by 13% overall and achieves a 9% improvement in pass@1, which is a substantial gain.
>
> > W3. The evaluation settings should be described in more detail, including specifics such as the hardware configurations. Additionally, the training overhead introduced by the proposed method, especially due to the iterative co-verification and co-evolution steps, should be reported and discussed to assess its practicality and scalability.
>
> A3. We have already reported the evaluation hardware setup in Appendix D.3: All executions in our experiments (e.g., co-verify and evaluation) are conducted on a CPU (Intel 535 i9-14900KF) and a GPU (RTX 4090 with 128 SMs). We use g++ 9.3.0 to compile C programs and nvcc 12.1 for CUDA programs. During execution, we set a maximum timeout of 60 seconds. Any program that exceeds this limit is treated as a runtime error. Thanks to the excellent open-source projects vLLM and LLaMA-Factory, as well as our multi-threaded optimizations, our framework achieves high training efficiency. For full fine-tuning of 7B/8B models, using 8×A100-80G GPUs with DeepSpeed ZeRO-3, we complete 4 training rounds to convergence in \~16 hours. Compared to standard fine-tuning, our framework introduces an additional Co-verify step in each round, which involves inference from both the Translator and the Tester (\~20 minutes per round), as well as execution-based verification (\~2 hours per round).
>
> During evaluation, we generate 20 samples for each of the 233 test cases and verify them in parallel, taking only \~1 hour.
>
> > Q1. The paper states that "while it is possible to train separate Translators and Testers for C and CUDA, we find that language-agnostic models perform better", and attributes this to the larger amount of training data available when combining multiple languages. Could the authors provide empirical evidence or quantitative results to support this claim? Are there quantitative results comparing language-specific and language-agnostic models? Additionally, if more training data from other languages were included, would the performance improve further?
>
> A4. In early experiments, we compared training two separate models with training a single language-agnostic model and found the language-agnostic model performed better. These results are from the first training round.
>
> |   |   |   |   |   |
> |---|---|---|---|---|
> ||BLEU|CodeBLEU|CPass|Pass@1|
> |sep: r1|73.33|74.91|89.96|74.66|
> |lang-agnostic: r1|79.24|78.72|91.84|76.85|
>
> We attribute this result to three main reasons:
>
> 1. C and CUDA share similar syntax and low-level semantics, enabling the model to benefit from joint training.
> 2. As two distinct programming paradigms, translating in both directions helps the model learn parallel programming concepts(e.g., how to convert `for` loops into CUDA thread operations and vice versa—which are mutually reinforcing tasks.)
> 3. Since the two directions are complementary, training a language-agnostic model effectively doubles the data, allowing the model to learn better representations.
>
> We mixed 568 parallel Python/Java samples collected from UniTrans into our round2 training data and trained using the same hyper-parameters as in the paper. The results are as follows:
>
> |   |   |   |   |   |   |   |
> |---|---|---|---|---|---|---|
> ||BLEU|CodeBLEU|CPass|Pass@1|Pass@5|Pass@10|
> |round2|84.24|84.32|95.28|84.08|93.48|94.61|
> |round2 w/ Java/Python|83.45|83.37|94.84|88.75|95.62|96.49|
>
> Experiments show that BLEU, CodeBLEU, and CPass scores drop, while correctness metrics like pass@k improve. This indicates that incorporating additional parallel corpora affects the model's ability to generate C/CUDA-style code, but also suggests that translation between other languages can enhance the model's C-to-CUDA translation capability.

---

> > ### Comment · Reviewer_bgUV · 2025-08-05
> >
> > Thank you for the clarification. I have raised my rating accordingly.

---

> > > ### Author Response · Authors · 2025-08-06
> > >
> > > Thank you for taking the time to review our paper and for recognizing our efforts. Your questions raise important points, and we will incorporate these clarifications into the final version. We sincerely appreciate your feedback and support.

---

### Note · Authors · 2025-08-14

Dear Chair and Reviewers,

We sincerely thank the Area Chair and reviewers for their thorough evaluation and insightful comments that improved our paper.

We are delighted that the reviewers have recognized the MuSL in our work as "theoretically elegant, novel and clever combination of existing techniques and a creative solution to the data scarcity problem" (Ekfk), and they concur that our work holds "important to hpc and represents a meaningful contribution" (bgUV, Ekfk, YS4J).

The main concerns raised by the reviewers center on the quality of the tests generated by the Tester (Rmzg, Ekfk), the computational cost (bgUV, Ekfk), and comparison with other iterative methods (YS4J).

In our rebuttal, we evaluated the coverage of the tests generated by the Tester and compare with the search-based EvoSuite, indicating that our LLM-based Tester can achieve high coverage and better capture the computational logic of C/CUDA, thereby producing more numerically sensitive tests. We have also provided detailed computational cost analysis, along with comparisons to existing iterative frameworks, highlighting the novelty of our approach.

Reviewer bgUV inquired about language-agnostic and suggested incorporating additional languages into training. We clarified the concept, addressed C/CUDA-specific challenges through our framework, and observed that adding Python/Java further improved pass@1, demonstrating the applicability of our framework.

Reviewer Rmzg inquired about how C/CUDA outputs are compared. We provided illustrative examples, effectively addressing the reviewer's concerns.

Reviewer Ekfk raised questions about the generalization of Tester training. We clarified that our primary focus is on improving the Tester's ability to generate valid inputs which is a challenging problem for C/CUDA, thereby validating the significance of the MuSL framework.

Reviewers Rmzg and YS4J initially gave a score of 4 and provide follow-up responses. Reviewer YS4J stated, "Thank you for addressing my questions; I remain positive about the paper." and reviewer Rmzg commented, "Thank you, please add these details to the paper." indicating that their concerns have been resolved. Reviewers bgUV and Ekfk initially assigned a score of 3, but after our clarifications, both increased their scores, reflecting their recognition of our work.

We sincerely thank the reviewers for their constructive feedback and the Area Chair for leading the discussions.


Best regrads,

The Authors

---

### Decision · Program_Chairs · 2025-09-17

**Decision:**

Accept (poster)

**Comment:**

This paper proposes MuSL, a mutually supervised Translator–Tester loop for sequential to parallel code translation that generates and verifies high‑coverage test cases, yielding substantial gains. Test quality, coverage, efficiency, and scalability concerns were addressed convincingly. Reviewers converged to positive scores. The AC recommends Acceptance and the final decision.